# EMaQ: Expected-Max Q-Learning Operator for Simple Yet Effective Offline and Online RL

## Abstract

Off-policy reinforcement learning (RL) holds the promise of sample-efficient learning of decision-making policies by leveraging past experience. However, in the offline RL setting – where a fixed collection of interactions are provided and no further interactions are allowed – it has been shown that standard off-policy RL methods can significantly underperform. Recently proposed methods often aim to address this shortcoming by constraining learned policies to remain close to the given dataset of interactions. In this work, we closely investigate an important simplification of BCQ (Fujimoto et al., 2018a) – a prior approach for offline RL – which removes a heuristic design choice and naturally restrict extracted policies to remain *exactly* within the support of a given behavior policy. Importantly, in contrast to their original theoretical considerations, we derive this simplified algorithm through the introduction of a novel backup operator, Expected-Max Q-Learning (EMaQ), which is more closely related to the resulting practical algorithm. Specifically, in addition to the distribution support, EMaQ explicitly considers the number of samples and the proposal distribution, allowing us to derive new sub-optimality bounds which can serve as a novel measure of complexity for offline RL problems. In the offline RL setting – the main focus of this work – EMaQ matches and outperforms prior state-of-the-art in the D4RL benchmarks (Fu et al., 2020a). In the online RL setting, we demonstrate that EMaQ is competitive with Soft Actor Critic (SAC). The key contributions of our empirical findings are demonstrating the importance of careful generative model design for estimating behavior policies, and an intuitive notion of complexity for offline RL problems. With its simple interpretation and fewer moving parts, such as no explicit function approximator representing the policy, EMaQ serves as a strong yet easy to implement baseline for future work.

## 1 Introduction

Leveraging past interactions in order to improve a decision-making process is the hallmark goal of off-policy reinforcement learning (RL) (Precup et al., 2001; Degris et al., 2012). Effectively learning from past experiences can significantly reduce the amount of online interaction required to learn a good policy, and is a particularly crucial ingredient in settings where interactions are costly or safety is of great importance, such as robotics Gu et al. (2017); Kalashnikov et al. (2018a), health Murphy et al. (2001), dialog agents (Jaques et al., 2019), and education Mandel et al. (2014). In recent years, with neural networks taking a more central role in the RL literature, there have been significant advances in developing off-policy RL algorithms for the function approximator setting, where policies and value functions are represented by neural networks (Mnih et al., 2015; Lillicrap et al., 2015; Gu et al., 2016b;a; Haarnoja et al., 2018; Fujimoto et al., 2018b). Such algorithms, while off-policy in nature, are typically trained in an online setting where algorithm updates are interleaved with additional online interactions. However, in purely offline RL settings, where a dataset of interactions are provided ahead of time and no additional interactions are allowed, the performance of these algorithms degrades drastically (Fujimoto et al., 2018a; Jaques et al., 2019).

A number of recent methods have been developed to address this shortcoming of off-policy RL algorithms. A particular class of algorithms for offline RL that have enjoyed recent success are

those based on dynamic programming and value estimation (Fujimoto et al., 2018a; Jaques et al., 2019; Kumar et al., 2019; Wu et al., 2019; Levine et al., 2020). Most proposed algorithms are designed with a key intuition that it is desirable to prevent policies from deviating too much from the provided collection of interactions. By moving far from the actions taken in the offline data, any subsequently learned policies or value functions may not generalize well and lead to the belief that certain actions will lead to better outcomes than they actually would. Furthermore, due to the dynamics of the MDP, taking out-of-distribution actions may lead to states not covered in the offline data, creating a snowball effect (Ross et al., 2011). In order to prevent learned policies from straying from the offline data, various methods have been introduced for regularizing the policy towards a base behavior policy (e.g. through a divergence penalty (Jaques et al., 2019; Wu et al., 2019; Kumar et al., 2019) or clipping actions (Fujimoto et al., 2018a)).

Taking the above intuitions into consideration, in this work we investigate a simplifcation of the BCQ algorithm (Fujimoto et al., 2018a) (a notable prior work in offline RL), which removes a heuristic design choice and has the property that extracted policies remain exactly within the support of a given behavior policy. In contrast to the theoretical considerations in the original work, we derive this simplified algorithm in a theoretical setup that more closely reflects the resulting algorithm. We introduce the Expected-Max Q-Learning (EMaQ) operator, which interpolates between the standard Q-function evaluation and Q-learning backup operators. The EMaQ operator makes explicit the relation between the proposal distribution and number of samples used, and leads to sub-optimality bounds which introduce a novel notion of complexity for offline RL problems. In its practical implementation for the continuous control and function approximator setting, EMaQ has only two standard components (an estimate of the base behavior policy, and Q functions) and does not explicitly represent a policy, requiring fitting one less function approximator than prior approaches (Fujimoto et al., 2018a; Kumar et al., 2019; Wu et al., 2019).

In online RL, EMaQ is competitive with Soft Actor Critic (SAC) (Haarnoja et al., 2018) and surpasses SAC in the deployment-efficient setting (Matsushima et al., 2020). In the offline RL setting – the main focus of this work – EMaQ matches and outperforms prior state-of-the-art in the D4RL (Fu et al., 2020a) benchmark tasks. Through our explorations with EMaQ we make two intriguing findings. First, due to the strong dependence of EMaQ on the quality of behavior policy used, our results demonstrate the significant impact of careful considerations in modeling the behavior policy that generate the offline interaction datasets. Second, relating to the introduced notion of complexity, in a diverse array of benchmark settings considered in this work we observe that surprisingly little modification to a base behavior policy is necessary to obtain a performant policy. As an example, we observe that while a HalfCheetah random policy obtains a return of 0, a policy that at each state uniformly samples only 5 actions and chooses the one with the best value obtains a return of 2000. The simplicity, intuitive interpretation, and strong empirical performance of EMaQ make it a great test-bed for further examination and theoretical analyses, and an easy to implement yet strong baseline for future work in offline RL.

## 2 BACKGROUND

Throughout this work, we represent Markov Decision Process (MDP) as $M = \langle \mathcal{S}, \mathcal{A}, r, \mathcal{P}, \gamma \rangle$, with state space $\mathcal{S}$, action space $\mathcal{A}$, reward function $r : \mathcal{S} \times \mathcal{A} \to \mathbb{R}$, transition dynamics $\mathcal{P}$, and discount $\gamma$. In offline RL, we assume access to a dataset of interactions with the MDP, which we will represent as collection of tuples $D = \{(s, a, s', r, t)\}^N$, where $t$ is an indicator variable that is set to True when $s'$ is a terminal state. We will use $\mu$ to represent the behavior policy used to collect $D$, and depending on the context, we will overload this notation and use $\mu$ to represent an estimate of the true behavior policy. For a given policy $\pi$, we will use the notation $d^\pi(s), d^\pi(s, a)$ to represent the state-visitation and state-action visitation distributions respectively.

As alluded to above, a significant challenge of offline RL methods is the problem of distribution shift. At training-time, there is no distribution shift in states as a fixed dataset $D$ is used for training, and the policy and value functions are never evaluated on states outside of $d^\mu(s)$. However, a very significant challenge is the problem of **distribution shift in actions**. Consider the Bellman backup for obtaining the Q-function of a given policy $\pi$,

$$\mathcal{T}_\pi Q(s, a) := r(s, a) + \gamma \cdot \mathbb{E}_{s'} \mathbb{E}_{a' \sim \pi(a'|s')} \Big[ Q(s', a') \Big] \tag{1}$$

The target Q-values on the right hand side depend on action samples $a' \sim \pi(a'|s')$. If the sampled actions are outside the distribution of actions observed in $D$, the estimated Q-values can be erroneous leading to incorrect target values. The effects of action distribution shift are further exacerbated in actor-critic algorithms; out of distribution (OOD) actions may incorrectly be assigned high values, in which case the policy will be updated to further sample OOD actions, leading to a hazardous loop.

An important approach – with particular recent interest – to mitigate the effects of both kinds of distributional shift is to devise methods for constraining learned policies to remain close to the behavior policy $\mu$: $d^\pi(s,a) \approx d^\mu(s,a)$. Below, we set the stage by reviewing a closely related prior work in offline RL.

**Batch Constrained Q-Learning (BCQ)**    In BCQ (Fujimoto et al., 2018a) the aim is to constrain a Q-Learning based algorithm such that it will be effective in the offline RL continuous control setting with function approximators. To do so, the trained policy is parameterized as:

$$\pi_\theta(a|s) = \underset{a_i + \xi_\theta(s, a_i)}{\arg\max} \ Q_\psi(s, a_i + \xi_\theta(s, a_i)) \qquad \text{for} \qquad a_i \sim \mu(a|s), i = 1, ..., N \qquad (2)$$

$$y(s, a, s', r, t) = \left( r + (1-t) \cdot \gamma \max_{a_i'} Q_{\psi'}(s', a_i') \right) \qquad \text{for} \qquad a_i' \sim \pi_\theta(a'|s'), i = 1, ..., N \quad (3)$$

$$\mathcal{L}_Q = \left( y(s, a, s', r, t) - Q_\psi(s, a) \right)^2 \qquad (4)$$

where $y(s, a)$ are target Q-values, $Q_\psi$ is learned with the objective in equation 4, $\mu(a|s)$ is an estimate of the base behavior policy (a generative model trained using the dataset $D$), and $\xi_\theta$ is an action perturbation model trained to modify actions towards more optimal ones. *Crucially*, each component of the output of $\xi_\theta$ is bounded to the range $[-\Phi, \Phi]$. ***The key intuition*** is that because $a_i$ are sampled from an estimate of the behavior policy, they should hopefully be within the distribution observed in $D$. Thus, since the perturbation model is constrained by the hyperparameter $\Phi$, the perturbed actions should not be too far from actions in the dataset. This should mitigate errors in value estimates, which should in turn lead to better updates for the perturbation model.

## 3    EXPECTED-MAX Q-LEARNING

We make the observation that, in the BCQ algorithm, if we could obtain a good estimate $\mu(a|s)$ and sufficiently increased the number of samples $N$, there would be no need for the perturbation network $\xi_\theta$. This simplification would remove one additional function approximator and the associated hyperparameter $\Phi$. This is the driving intuition of our work, which we frame theoretically in a manner that encapsulates the key components: the behavior policy $\mu$ and number of samples $N$. Below, we introduce the Expected-Max Q operator, illustrate its key properties for tabular MDPs, and obtain sub-optimality bounds which can serve as a novel measure of complexity of an offline RL problem for future theoretical work. We then provide an extension to the offline RL setting with function approximators, and then discuss the generative model used to approximate the behavior policy.

### 3.1    EXPECTED-MAX Q OPERATOR

Let $\mu(a|s)$ be an arbitrary behavior policy, and let $\{a_i\}^N \sim \mu(a|s)$ denote sampling $N$ iid actions from $\mu(a|s)$. Let $Q : \mathcal{S} \times \mathcal{A} \to \mathbb{R}$ be an arbitrary function. For a given choice of $N$, we define the Expected-Max Q-Learning operator (EMaQ) $\mathcal{T}_\mu^N Q$ as follows:

$$\mathcal{T}_\mu^N Q(s, a) := r(s, a) + \gamma \cdot \mathbb{E}_{s'} \mathbb{E}_{\{a_i'\}^N \sim \mu(\cdot|s')} \left[ \max_{a' \in \{a_i'\}^N} Q(s', a') \right] \quad (5)$$

$$\text{Q-Evaluation for } \mu \qquad \mathcal{T}_\mu Q(s, a) := r(s, a) + \gamma \cdot \mathbb{E}_{s'} \mathbb{E}_{a' \sim \mu(\cdot|s')} \left[ Q(s', a') \right] \qquad (6)$$

$$\text{Q-Learning} \qquad \mathcal{T}^* Q(s, a) := r(s, a) + \gamma \cdot \mathbb{E}_{s'} \left[ \max_{a'} Q(s', a') \right] \qquad (7)$$

This operator provides a natural interpolant between the on-policy backup for $\mu$ (Eq. 6) when $N = 1$, and the Q-learning backup (Eq. 7) as $N \to \infty$ (if $\mu(a|s)$ has full support over $\mathcal{A}$). We formalize these observations more precisely below when we articulate the key properties in the tabular MDP setting. We discuss how this relates to existing modified backup operators in the related work.

### 3.2 Dynamic Programming Properties in the Tabular MDP Setting

To understand any novel backup operator it is useful to first characterize its key dynamic programming properties in the tabular MDP setting. First, we establish that EMaQ retains essential contraction and fixed-point existence properties, regardless of the choice of $N \in \mathbb{N}$. In the interest of space, all missing proofs can be found in Appendix A.

**Theorem 3.1.** *In the tabular setting, for any $N \in \mathbb{N}$, $\mathcal{T}_\mu^N$ is a contraction operator in the $\mathcal{L}_\infty$ norm. Hence, with repeated applications of the $\mathcal{T}_\mu^N$, any initial $Q$ function converges to a unique fixed point.*

**Theorem 3.2.** *Let $Q_\mu^N$ denote the unique fixed point achieved in Theorem 3.1, and let $\pi_\mu^N(a|s)$ denote the policy that samples $N$ actions from $\mu(a|s)$, $\{a_i\}^N$, and chooses the action with the maximum $Q_\mu^N$. Then $Q_\mu^N$ is the Q-value function corresponding to $\pi_\mu^N(a|s)$.*

*Proof. (Theorem 3.2)* Rearranging the terms in equation 5 we have,

$$\mathcal{T}_\mu^N Q_\mu^N(s,a) = r(s,a) + \gamma \cdot \mathbb{E}_{s'} \mathbb{E}_{a' \sim \pi_\mu^N(a'|s')}[Q_\mu^N(s',a')]$$

Since by definition $Q_\mu^N$ is the unique fixed point of $\mathcal{T}_\mu^N$, we have our result. □

From these results we can then rigorously establish the interpolation properties of the EMaQ family.

**Theorem 3.3.** *Let $\pi_\mu^*$ denote the optimal policy from the class of policies whose actions are restricted to lie within the support of the policy $\mu(a|s)$. Let $Q_\mu^*$ denote the Q-value function corresponding to $\pi_\mu^*$. Furthermore, let $Q_\mu$ denote the Q-value function of the policy $\mu(a|s)$. Let $\mu^*(s) := \int_{Support(\pi_\mu^*(a|s))} \mu(a|s)$ denote the probability of optimal actions under $\mu(a|s)$. Under the assumption that $\inf_s \mu^*(s) > 0$ and $r(s,a)$ is bounded, we have that,*

$$Q_\mu^1 = Q_\mu \qquad and \qquad \lim_{N \to \infty} Q_\mu^N = Q_\mu^*$$

That is, Theorem 3.3 shows that, given a base behavior policy $\mu(a|s)$, the choice of $N$ makes the EMaQ operator interpolate between evaluating the Q-value of $\mu$ on the one hand, and learning the optimal Q-value function on the other (optimal subject to the support constraint discussed in Theorem 3.3). In the special case where $\mu(a|s)$ has full support over the action space $\mathcal{A}$, EMaQ interpolates between the standard Q-Evaluation and Q-Learning operators in reinforcement learning.

Intuitively, as we increase $N$, the fixed-points $Q_\mu^N$ should correspond to increasingly better policies $\pi_\mu^N(a|s)$. We show that this is indeed the case.

**Theorem 3.4.** *For all $N, M \in \mathbb{N}$, where $N > M$, we have that $\forall s \in \mathcal{S}, \forall a \in \text{Support}(\mu(\cdot|s))$, $Q_\mu^N(s,a) \geq Q_\mu^M(s,a)$. Hence, $\pi_\mu^N(a|s)$ is at least as good of a policy as $\pi_\mu^M(a|s)$.*

It is also valuable to obtain a sense of how suboptimal $\pi_\mu^N(a|s)$ may be with respect to the optimal policy supported by the policy $\mu(a|s)$.

**Theorem 3.5.** *For $s \in \mathcal{S}$ let,*

$$\Delta(s,N) = \max_{a \in \text{Support}(\mu(\cdot|s))} Q_\mu^*(s,a) - \mathbb{E}_{\{a_i\}^N \sim \mu(\cdot|s)}\big[\max_{b \in \{a_i\}^N} Q_\mu^*(s,b)\big]$$

*The suboptimality of $Q_\mu^N$ can be upperbounded as follows,*

$$\left\| Q_\mu^N - Q_\mu^* \right\|_\infty \leq \frac{\gamma}{1-\gamma} \max_{s,a} \mathbb{E}_{s'} \Big[ \Delta(s',N) \Big] \leq \frac{\gamma}{1-\gamma} \max_s \Delta(s,N) \qquad (8)$$

*The same also holds when $Q_\mu^*$ is replaced with $Q_\mu^N$ in the definition of $\Delta$.*

### 3.3 A Measure of Complexity for Offline RL

The bounds in equation 8 capture the main intuitions about the interplay between $\mu(a|s)$ and the choice of $N$. If for each state, $\mu(a|s)$ places sufficient mass over the optimal actions, $\pi_\mu^N$ will be

---

**Algorithm 1:** Test-Time Policy $\pi_{\text{test}}$

---

**Function** `TestEnsemble`(*values*)**:**
   |   **return** $\lambda \cdot \min(values) + (1 - \lambda) \cdot \max(values)$
**Function** $\pi_{\text{test}}$ ($s$)**:**
   |   $\{a_i\}^N \sim \mu(a|s)$
   |   **return** $\arg\max_{\{a_i\}^N}$ `TestEnsemble`$\big([Q_i(s, a)$ for all $i]\big)$

---

close to $\pi_{\mu}^*$. The two variants of $\Delta(s, N)$, based on $Q_{\mu}^*$ or $Q_{\mu}^N$, suggest an intriguing notion of difficulty for an offline RL problem. If we could estimate either of these Q-value functions, then for a desired set of states (such as initial states) we could plot $\Delta(s, N)$ as decreasing function of $N$. The rate at which this function decreases could serve as an intuitive notion of difficulty for a given offline offline RL problem which consists of an MDP and a given behavior policy. While we leave theoretical investigations of this measure for future work, our empirical results in Section 5 demonstrate that the effective value of $N$ may be surprisingly small.

### 3.4 OFFLINE RL SETTING WITH FUNCTION APPROXIMATORS

Typically, we are not provided with the policies that generated the provided trajectories. Hence, as a first step we fit a generative model $\mu(a|s)$ to the $(s, a)$ pairs in the offline dataset, representing the mixture of policies that generated this data (details below). Having obtained $\mu(a|s)$, we move on to the EMaQ training procedure. Similar to prior works (Fujimoto et al., 2018a; Kumar et al., 2019; Wu et al., 2019), we train $K$ Q functions (represented by MLPs) and make use of an ensembling procedure to combat overestimation bias (Hasselt, 2010; Van Hasselt et al., 2016; Fujimoto et al., 2018b). Letting $D$ represent the offline dataset, the objective for the Q functions takes the following form:

$$\mathcal{L}(\theta_i) = \mathbb{E}_{(s,a,s',r,t)\sim D}\left[\left(Q_{\theta_i}(s, a) - y(s, a, s', r, t)\right)^2\right] \tag{9}$$

$$y(s, a, s', r, t) = \left(r + (1 - t) \cdot \gamma \max_{a'_i} Q'_{ens}(s', a'_i)\right) \quad \text{for} \quad a'_i \sim \mu(a'|s'), i = 1, ..., N \tag{10}$$

where $t$ is the indicator variable $\mathbb{1}[s'$ is terminal], and $Q'_{ens}$ represents the ensemble of target Q functions. In short, we sample $N$ actions from $\mu(a'|s')$ and take the value of the best action to form the target. The algorithm box describing the full training loop can be viewed in Algorithm 2.

**Notably, we do not train an explicit neural network representing the policy.** At test-time, given a state $s$, we sample $N$ actions from $\mu(a|s)$ and choose the action with the maximum value under the ensemble of Q functions (see Algorithm 1). While `TestEnsemble` can differ from the `Ensemble` function used to compute target Q values[1], in this work we used the same ensembling procedure with $\lambda = 1.0$ (with the exception of experiments in Section F).

### 3.5 INTUITION FOR OFFLINE EMAQ AND MODELING $\mu(a|s)$

The intuition for how offline EMaQ aims to address the problem of erroneous value estimates can be understood from attending to equation 10 and the form of the test-time policy (Algorithm 1). In equation 10 we observe that target values for the Q functions are computed by sampling actions **from the behavior policy estimate $\mu$**, and not a separately learned policy that may sample out of distribution actions, as in BCQ. At test-time, our implicit policy is also formed by choosing amongst actions sampled from $\mu$. Hence, if $\mu$ accurately estimates the behavior policy well, we will never sample actions outside the support and will not need to evaluate the value of such actions. In the practical setting where $\mu$ may have inaccuracies, the hyperparameter $N$ acts as an implicit regularizer: Using very large values of $N$ maximizes the chance of sampling actions that are out

---

[1] some examples of alternative choices are `mean`, `max`, UCB-style estimates, or simply using just one of the trained Q functions

of distribution and have erroneous value estimates, while smaller $N$ reduces the chance of this happening in every update iteration and therefore smoothens the incorrect values.

With the importance of a good behavior estimates accentuated in our proposed method, we pay closer attention to the choice of generative model used for representing $\mu$. Past works (Fujimoto et al., 2018a; Kumar et al., 2019; Wu et al., 2019) have typically used Variational Auto-Encoders (VAEs) (Kingma & Welling, 2013; Rezende et al., 2014) to represent the behavior distribution $\mu(a|s)$. Unfortunately, after training the aggregate posterior $q_{\text{agg}}(z) := \mathbb{E}_x[q(z|x)]$ of a VAE does not typically align well with its prior, making it challenging to sample from in a manner that effectively covers the distribution it was trained on[2]. We opt for using an autoregressive architecture based on MADE (Germain et al., 2015) as it allows for representing more expressive distributions and enables more accurate sampling. Inspired by recent works (Metz et al., 2017; Van de Wiele et al., 2020), our generative model architecture also makes use of discretization in each action dimension. Full details can be found in Appendix B.

## 4 RELATED WORK

**Offline RL**   Many recent methods for offline RL (Fujimoto et al., 2018a; Kumar et al., 2019; Wu et al., 2019; Jaques et al., 2019), where no interactive data collection is allowed during training, mostly rely on constraining the learned policy to stay close to the data collection distribution. Fujimoto et al. (2018a) clip the maximum deviation from actions sampled from a base behavior policy, while Kumar et al. (2019); Wu et al. (2019); Jaques et al. (2019) incorporate additional distributional penalties (such as KL divergence or MMD) for regularizing learned policies to remain close to the base policy. Our work is an instance of this family of approaches for offline RL; however, arguably our method is simpler as it does not involve learning an additional proposal-modifying policy Fujimoto et al. (2018a), or modifying reward functions (Kumar et al., 2019; Jaques et al., 2019).

**Finding Maximizing Actions**   Naïvely, EMaQ can also be seen as just performing approximate search for $\max_a Q(s, a)$ in standard Q-learning operator, which has been studied in various prior works for Q-learning in large scale spaces (e.g. continuous). NAF (Gu et al., 2016b) and ICNN (Amos et al., 2017) directly constrain the function family of Q-functions such that the optimization can be closed-form or tractable. QT-OPT (Kalashnikov et al., 2018b) makes use of two iterations of the Cross-Entropy Method (Rubinstein & Kroese, 2013), while CAQL (Ryu et al., 2019) uses Mixed-Integer Programming to find the exact maximizing action while also introducing faster approximate alternatives. In (Van de Wiele et al., 2020) – the most similar approach to our proposed method EMaQ – throughout training a mixture of uniform and learned proposal distributions are used to sample actions. The sampled actions are then evaluated under the learned Q functions, and the top K maximizing actions are distilled back into the proposal distribution. In contrast to our work, these works assume these are approximate maximization procedures and do not provide extensive analysis for the resulting TD operators. Our theoretical analysis on the family of TD operators described by EMaQ can therefore provide new perspectives on some of these highly successful Q-learning algorithms (Kalashnikov et al., 2018a; Van de Wiele et al., 2020) – particularly on how the proposal distribution affects convergence.

**Modified Backup Operators**   Many prior works study modifications to standard backup operators to achieve different convergence properties for action-value functions or their induced optimal policies. $\Psi$-learning (Rawlik et al., 2013) proposes a modified operator that corresponds to policy iterations with KL-constrained updates (Kakade, 2002; Peters et al., 2010; Schulman et al., 2015) where the action-value function converges to negative infinity for all sub-optimal actions. Similarly but distinctly, Fox et al. (2015); Jaques et al. (2017); Haarnoja et al. (2018); Nachum et al. (2017) study smoothed TD operators for a modified entropy- or KL-regularized RL objective. Bellemare et al. (2016) derives a family of consistent Bellman operators and shows that they lead to increasing action gaps (Farahmand, 2011) for more stable learning. However, most of these operators have not been studied in offline learning. Our work adds a novel family operators to this rich literature of

---

[2]past works typically clip the range of the latent variable $z$ and adjust the weighting of the KL term in the evidence lower-bound to ameliorate the situation

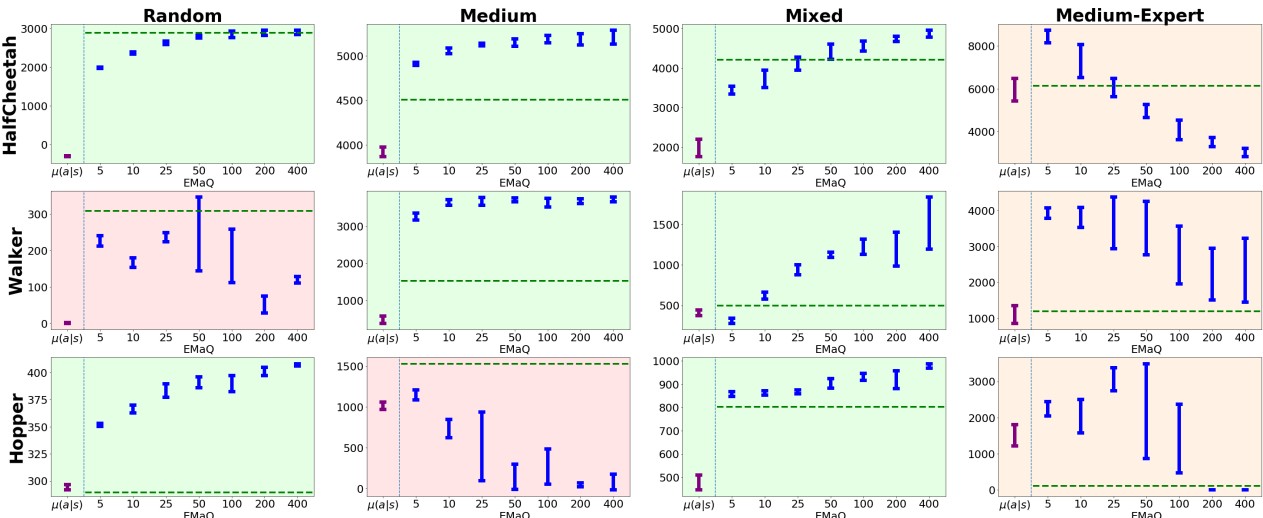

Figure 1: Results for evaluating EMaQ on D4RL benchmark's (Fu et al., 2020b) standard Mujoco domains, with $N \in \{5, 10, 25, 50, 100, 200, 400\}$. Values above $\mu(a|s)$ represent the result of evaluating the base behavior policies. Horizontal green lines represent the reported performance of BEAR in the D4RL benchmark (apples to apples comparisons in Figure 2). The types of offline datasets are: **random**: 1M transitions are collected by a random agent, **medium**: 1M transitions are collected by a half-trained SAC (Haarnoja et al., 2018) policy, **mixed**: the replay buffer of this half-trained policy, and **medium-expert**: combination of the medium dataset and 1M additional transitions from a fully trained policy. Refer to main text (Section 5.1) for description of color-coding. For better legibility, we have included a larger variant of these plots in the Appendix L. Full experimental details in Appendix G.

operators for RL, and provides strong empirical validation on how simple modifications of operators can translate to effective offline RL with function approximations.

## 5 EXPERIMENTS

For all experiments we make use of the codebase of (Wu et al., 2019), which presents the BRAC off-policy algorithm and examines the importance of various factors in BCQ (Fujimoto et al., 2018a) and BEAR (Kumar et al., 2019) methods. We implement EMaQ into this codebase. We make use of the recently proposed D4RL (Fu et al., 2020b) datasets for bechmarking offline RL.

**Online EMaQ** Despite obtaining **strong online RL results competitive with and outperforming SAC** (Haarnoja et al., 2018) (Figure 3), as the main focus of our work is for the offline setting, we have placed our online RL methodology and results in Appendix F. However, we emphasize that significance of obtaining strong online RL performance with effectively the same algorithm as the offline setting should not be overlooked. Most prior offline RL works have not considered how their methods might transfer to online or batched online setting, and recent work (Nair et al., 2020) has demonstrated the challenges of finetuning from a policy trained offline, in the online setting.

### 5.1 PRACTICAL EFFECT OF $N$ AND THE CHOICE OF GENERATIVE MODEL $\mu$

We begin by empirically evaluating key aspects of offline EMaQ, namely the effect of $N$, and choice of generative model used for representing the behavior estimate $\mu$. In prior approaches such as those described in the background section of this work, care must be taken in choosing the hyperparameter that dictates the extent to which learned policies can deviate from the base behavior policies; too small and we cannot improve upon the base policy, too large and the value of actions cannot be correctly estimated. In EMaQ, at least in theory, choosing higher values of $N$ should result in strictly better policies. Additionally, there exists a concern that $N$ may need to be impractically large. Thus, we empirically investigate to what extent the monotonic trend holds in practice, and seek to understand what magnitudes of $N$ result in good policies in practical benchmark domains. Figure 1 presents our results with $N \in \{5, 10, 25, 50, 100, 200, 400\}$. In the green plots, we observe that empirical results follow our intuitions: with increasing $N$ the resultant policies become

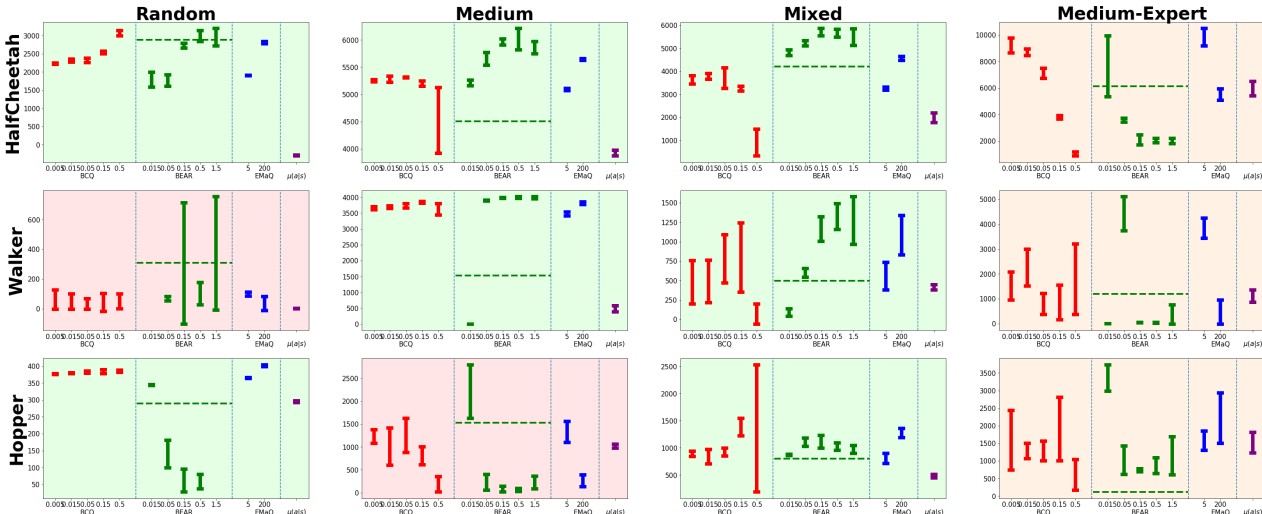

Figure 2: Comparison of EMaQ, BCQ, and BEAR on D4RL (Fu et al., 2020b) benchmark domains when using our proposed autoregressive $\mu(a|s)$. For both BCQ and BEAR, from left to right as the value of the hyperparameter increases, the allowed deviation from $\mu(a|s)$ increases. Horizontal green lines represent the reported performance of BEAR in the D4RL benchmark. Color-coding follows Figure 1. For better legibility, we have included a larger variant of these plots in the Appendix L. Full experimental details in Appendix G.

better. In the medium-expert settings (i.e. orange plots), while for smaller values of $N$ we observe strong performance, there appears to be a downward trend. As discussed in Section 3.5, smaller values of $N$ result in an implicit regularization. Hence, the orange plots may indicate that even with the stronger choice of generative models in EMaQ, inaccuracies in value estimates may still exist, suggesting the need for future work that introduces better regularizers for the value functions than ensembling (Kumar et al., 2020). Lastly, the red plots indicate settings where behavior is erratic. Closer examination of training curves and our experiments with other off-policy methods (Figure 2) suggests that this may be due to the intrinsic nature of these environment and data settings.

The dashed horizontal lines in Figure 2 represent the performance of BEAR – which uses a VAE for representing $\mu$ – as reported in the D4RL (Fu et al., 2020b) benchmark paper (apples to apples comparison in Section 5.2). Our results demonstrate that the combination of a strong generative model and EMaQ's simply constrained backup operator can match and in many cases noticeably exceed results from prior algorithms and design choices. Comparing Figure 2 to Figure 6 in the Appendix, **we observe that our choice of generative model is crucial to the performance of EMaQ**. With a VAE architecture as used in prior work, EMaQ's performance is significantly reduced, in most cases worse than prior reported results for BEAR, and never exhibits a monotonic trend as a function of $N$. This is despite the fact that when evaluating the performance of the behavior estimate $\mu$ under the two architecture choices results in almost identical results (the first column of each sub-plot corresponding to $\mu(a|s)$). We do not believe that autoregressive models are intrinsically better than VAEs, but rather our results demonstrate the need for more careful attention on the choice of $\mu(a|s)$. **Since EMaQ is closely tied to the choice of behavior model, it may be more effective for evaluating how well $\mu(a|s)$ represents the given offline dataset. From a practical perspective, our results suggest that for a given domain, focusing efforts on building-in good inductive biases in the generative models and value functions might be sufficient to obtain strong offline RL performance in many domains.**

## 5.2 COMPARISON ON D4RL OFFLINE RL BENCHMARK

To evaluate EMaQ with respect to prior methods, we compare to two popular and closely related prior methods for offline RL, BCQ (Fujimoto et al., 2018a) and BEAR (Kumar et al., 2019). As with the previous section, full experimental details can be found in Appendix G.1. Figure 2 and Table 1 present our empirical results. Note that with our proposed autoregressive models, the results for BEAR are matched and in some cases noticeably above the values reported in the D4RL benchmark (Fu et al., 2020b) (green horizontal lines). For easier interpretation, the plots are colored

| Setting | BC | BCQ | BEAR | EMaQ | EMaQ N |
|---|---|---|---|---|---|
| kitchen-complete | $27.2 \pm 3.2$ | $26.5 \pm 4.8$ | — | $\mathbf{36.9 \pm 3.7}$ | 64 |
| kitchen-partial | $46.2 \pm 2.8$ | $69.3 \pm 5.2$ | — | $\mathbf{74.6 \pm 0.6}$ | 8 |
| kitchen-mixed | $52.5 \pm 3.8$ | $65.5 \pm 1.8$ | — | $\mathbf{70.8 \pm 2.3}$ | 8 |
| antmaze-umaze | $59.0 \pm 5.5$ | $25.5 \pm 20.0$ | $56.3 \pm 28.8$ | $\mathbf{91.0 \pm 4.6}$ | 100 |
| antmaze-umaze-diverse | $58.8 \pm 9.5$ | $68.0 \pm 19.0$ | $57.5 \pm 39.2$ | $\mathbf{94.0 \pm 2.4}$ | 50 |
| antmaze-medium-play | $\mathbf{0.7 \pm 1.0}$ | $\mathbf{3.5 \pm 6.1}$ | $\mathbf{0.2 \pm 0.4}$ | $0.0 \pm 0.0$ | — |
| antmaze-medium-diverse | $\mathbf{0.4 \pm 0.8}$ | $\mathbf{0.5 \pm 0.9}$ | $\mathbf{0.2 \pm 0.4}$ | $0.0 \pm 0.0$ | — |
| antmaze-large-play | $0.0 \pm 0.0$ | $0.0 \pm 0.0$ | $0.0 \pm 0.0$ | $0.0 \pm 0.0$ | — |
| antmaze-large-diverse | $0.0 \pm 0.0$ | $0.0 \pm 0.0$ | $0.0 \pm 0.0$ | $0.0 \pm 0.0$ | — |
| door-cloned | $\mathbf{0.0 \pm 0.0}$ | $\mathbf{0.2 \pm 0.4}$ | $\mathbf{0.0 \pm 0.0}$ | $\mathbf{0.2 \pm 0.3}$ | 64 |
| hammer-cloned | $\mathbf{1.2 \pm 0.6}$ | $\mathbf{1.3 \pm 0.5}$ | $0.3 \pm 0.0$ | $1.0 \pm 0.7$ | 64 |
| pen-cloned | $24.5 \pm 10.2$ | $\mathbf{43.8 \pm 6.4}$ | $-3.1 \pm 0.2$ | $27.9 \pm 3.7$ | 128 |
| relocate-cloned | $\mathbf{-0.2 \pm 0.0}$ | $\mathbf{-0.2 \pm 0.0}$ | $\mathbf{0.0 \pm 0.0}$ | $\mathbf{-0.2 \pm 0.2}$ | 16 |

Table 1: Results on a series of other environments and data settings from the D4RL benchmark (Fu et al., 2020a). Results are normalized to the range $[0, 100]$, per the D4RL normalization scheme. For each method, for each environment and data setting the results of the best hyperparameter setting are reported. The last column indicates the best value of $N$ in EMaQ amongst the considered hyperparameters (for the larger `antmaze` domains, we do not report this value since no value of $N$ obtains nonzero returns). All the domains below the blue double-line are effectively unsolved by all methods. We have technical difficulties in evaluating BEAR on the kitchen domains. This manuscript will be updated upon obtaining these results. Additional details can be found in Appendix G.3.

the same as in Figure 1. Our key take-away is that despite its simplistic form, EMaQ is strongly competitive with prior state-of-the-art methods, and in the case of Table 1 outperforms prior approaches. Despite this, there remain many domains in the D4RL benchmark on which none of the considered algorithms make any progress (Table 1), indicating that much algorithmic advances are still necessary for solving many of the considered domains.

A very eye-catching result in above figures is that in almost all settings of the standard Mujoco environments (Figures 1 and 2), just $N = 5$ significantly improves upon $\mu(a|s)$ and in most settings matches or exceeds significantly beyond previously reported results. Concretely, this means that in the HalfCheetah-Random setting, if at each state we sample 5 actions uniformly random and choose the best one under the learned Q-value function, we convert a random policy with return 0 to a policy with return 2000. **In this way, EMaQ provides a quite intuitive and surprising measure of the complexity for offline RL problems. This empirical observation also corroborates our discussion in Section 3.3, encouraging future theoretical investigations into $\Delta(s, N)$.**

## 6 CONCLUSION

In this work, we investigate a significant simplification of the BCQ (Fujimoto et al., 2018a) algorithm by removing the heuristic perturbation network. By introducing the Expect-Max Q-Learning operator, we present a novel theoretical setup that takes into account the proposal distribution $\mu(a|s)$ and the number of action samples $N$, and hence more closely matches the resulting practical algorithm. With fewer moving parts and one less function approximator, EMaQ matches and outperforms prior state-of-the-art in online and offline RL. Our investigations with EMaQ demonstrate the significance of careful considerations in the design of generative models used. Furthermore, our theoretical and empirical findings bring into light novel notions of complexity for offline RL problems. Given the simplicity, tractable theory, and state-of-the-art performance of EMaQ, we hope our work can serve as a foundation for future works on understanding and improving offline RL.

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

## A PROOFS

All the provided proofs operate under the setting where $\mu(a|s)$ has full support over the action space. When this assumption is not satisfied, the provided proofs can be transferred by assuming we are operating in a new MDP $M_\mu$ as defined below.

Given the MDP $M = \langle S, A, r, P, \gamma \rangle$ and $\mu(a|s)$, let us define the new MDP $M_\mu = \langle S_\mu, A_\mu, r, P, \gamma \rangle$, where $S_\mu$ denotes the set of reachable states by $\mu$, and $A_\mu$ is $A$ restricted to the support of $\mu(a|s)$ in each state in $S_\mu$.

### A.1 CONTRACTION MAPPING

**Theorem 3.1.** *In the tabular setting, for any $N \in \mathbb{N}$, $\mathcal{T}_\mu^N$ is a contraction operator in the $\mathcal{L}_\infty$ norm. Hence, with repeated applications of the $\mathcal{T}_\mu^N$, any initial Q function converges to a unique fixed point.*

*Proof.* Let $Q_1$ and $Q_2$ be two arbitrary $Q$ functions.

$$\left\| \mathcal{T}_\mu^N Q_1 - \mathcal{T}_\mu^N Q_2 \right\|_\infty = \tag{11}$$

$$\max_{s,a} \left| \left( r(s,a) + \gamma \cdot \mathbb{E}_{s'} \mathbb{E}_{\{a_i\}^N} \left[ \max_{\{a_i\}^N} Q_1(s',a') \right] \right) - \left( r(s,a) + \gamma \cdot \mathbb{E}_{s'} \mathbb{E}_{\{a_i\}^N} \left[ \max_{\{a_i\}^N} Q_2(s',a') \right] \right) \right| = \tag{12}$$

$$\gamma \cdot \max_{s,a} \left| \mathbb{E}_{s'} \mathbb{E}_{\{a_i\}^N} \left[ \max_{\{a_i\}^N} Q_1(s',a') - \max_{\{a_i\}^N} Q_2(s',a') \right] \right| \leq \tag{13}$$

$$\gamma \cdot \max_{s,a} \mathbb{E}_{s'} \mathbb{E}_{\{a_i\}^N} \left| \max_{\{a_i\}^N} Q_1(s',a') - \max_{\{a_i\}^N} Q_2(s',a') \right| \leq \tag{14}$$

$$\gamma \cdot \max_{s,a} \mathbb{E}_{s'} \mathbb{E}_{\{a_i\}^N} \| Q_1 - Q_2 \|_\infty = \tag{15}$$

$$\gamma \cdot \| Q_1 - Q_2 \|_\infty \tag{16}$$

where line 15 is due to the following: Let $\hat{a} = \arg\max_{\{a_i\}^N} Q_1(s', a_i)$,

$$\max_{\{a_i\}^N} Q_1(s',a') - \max_{\{a_i\}^N} Q_2(s',a') = Q_1(s',\hat{a}) - \max_{\{a_i\}^N} Q_2(s',a') \tag{17}$$

$$\leq Q_1(s',\hat{a}) - Q_2(s',\hat{a}) \tag{18}$$

$$\leq \| Q_1 - Q_2 \|_\infty \tag{19}$$

$\square$

### A.2 LIMITING BEHAVIOR

**Theorem 3.3.** *Let $\pi_\mu^*$ denote the optimal policy from the class of policies whose actions are restricted to lie within the support of the policy $\mu(a|s)$. Let $Q_\mu^*$ denote the Q-value function corresponding to $\pi_\mu^*$. Furthermore, let $Q_\mu$ denote the Q-value function of the policy $\mu(a|s)$. Let $\mu^*(s) := \int_{Support(\pi_\mu^*(a|s))} \mu(a|s)$ denote the probability of optimal actions under $\mu(a|s)$. Under the assumption that $\inf_s \mu^*(s) > 0$ and $r(s,a)$, we have that,*

$$Q_\mu^1 = Q_\mu \qquad \text{and} \qquad \lim_{N \to \infty} Q_\mu^N = Q_\mu^*$$

Let $\mu^*(s) := \int_{\text{Support}(\pi_\mu^*(a|s))} \mu(a|s)$ denote the probability of optimal actions under $\mu(a|s)$. To show $\lim_{N \to \infty} Q_\mu^N = Q_\mu^*$, we also require the additional assumption that $\inf_s \mu^*(s) > 0$.

*Proof.* Given that,

$$\mathcal{T}_\mu^1 Q(s,a) := r(s,a) + \gamma \cdot \mathbb{E}_{s'} \mathbb{E}_{\{a_i\}^N \sim \mu(\cdot|s')} \left[ Q(s',a') \right] \tag{20}$$

the unique fixed-point of $\mathcal{T}_\mu^1$ is the Q-value function of the policy $\mu(a|s)$. Hence $Q_\mu^1 = Q_\mu$.

The second part of this theorem will be proven as a Corollary to Theorem 3.5

$\square$

## A.3 INCREASINGLY BETTER POLICIES

**Theorem 3.4.** *For all $N, M \in \mathbb{N}$, where $N > M$, we have that $\forall s \in \mathcal{S}, \forall a \in \text{Support}(\mu(\cdot|s))$, $Q_\mu^N(s,a) \geq Q_\mu^M(s,a)$. Hence, $\pi_\mu^N(a|s)$ is at least as good of a policy as $\pi_\mu^M(a|s)$.*

*Proof.* It is sufficient to show that $\forall s, a, Q_\mu^{N+1}(s,a) \geq Q_\mu^N(s,a)$. We will do so by induction. Let $Q^i$ denote the resulting function after applying $\mathcal{T}_\mu^{N+1}$, $i$ times, starting from $Q_\mu^N$.

Base Case

By definition $Q^0 := Q_\mu^N$. Let $s \in \mathcal{S}$, $a \in \mathcal{A}$.

$$Q^1(s,a) = \mathcal{T}_\mu^{N+1} Q^0(s,a) \tag{21}$$

$$= r(s,a) + \gamma \cdot \mathbb{E}_{s'} \mathbb{E}_{\{a_i\}^{N+1} \sim \mu(a'|s')} \big[ \max_{\{a_i\}^{N+1}} Q^0(s',a') \big] \tag{22}$$

$$\geq r(s,a) + \gamma \cdot \mathbb{E}_{s'} \mathbb{E}_{\{a_i\}^N \sim \mu(a'|s')} \big[ \max_{\{a_i\}^N} Q^0(s',a') \big] \tag{23}$$

$$= r(s,a) + \gamma \cdot \mathbb{E}_{s'} \mathbb{E}_{\{a_i\}^N \sim \mu(a'|s')} \big[ \max_{\{a_i\}^N} Q_\mu^N(s',a') \big] \tag{24}$$

$$= Q_\mu^N(s,a) \tag{25}$$

$$= Q^0(s,a) \tag{26}$$

Induction Step

Assume $\forall s, a, Q^i(s,a) \geq Q^{i-1}(s,a)$.

$$Q^{i+1}(s,a) - Q^i(s,a) = \mathcal{T}_\mu^{N+1} Q^i(s,a) - \mathcal{T}_\mu^{N+1} Q^{i-1}(s,a) \tag{27}$$

$$= \gamma \cdot \mathbb{E}_{s'} \mathbb{E}_{\{a_i\}^{N+1} \sim \mu(a'|s')} \big[ \max_{\{a_i\}^{N+1}} Q^i(s',a') - \max_{\{a_i\}^{N+1}} Q^{i-1}(s',a') \big] \tag{28}$$

$$\geq 0 \tag{29}$$

Hence, by induction we have to $\forall i, j, i > j \implies \forall s, a, Q^i(s,a) \geq Q^j(s,a)$. Since $Q^0 = Q_\mu^N$ and $\lim_{i \to \infty} Q^i = Q_\mu^{N+1}$, we have than $\forall s, a, Q_\mu^{N+1}(s,a) \geq Q_\mu^N(s,a)$. Thus $\pi_\mu^{N+1}$ is a better policy than $\pi_\mu^N$, and by a simple induction argument, $\pi_\mu^N$ is a better policy than $\pi_\mu^M$ when $N > M$.

$\square$

## A.4 BOUNDS

**Theorem 3.5.** *For $s \in \mathcal{S}$ let,*

$$\Delta(s) = \max_{a \in \text{Support}(\mu(\cdot|s))} Q_\mu^*(s,a) - \mathbb{E}_{\{a_i\}^N \sim \mu(\cdot|s)} \big[ \max_{b \in \{a_i\}^N} Q_\mu^*(s,b) \big]$$

*The suboptimality of $Q_\mu^N$ can be upperbounded as follows,*

$$\big\| Q_\mu^N - Q_\mu^* \big\|_\infty \leq \frac{\gamma}{1-\gamma} \max_{s,a} \mathbb{E}_{s'} \big[ \Delta(s') \big] \leq \frac{\gamma}{1-\gamma} \max_s \Delta(s) \tag{30}$$

*The same also holds when $Q_\mu^*$ is replaced with $Q_\mu^N$ in the definition of $\Delta$.*

*Proof.* The two versions where $\Delta(s)$ is defined in terms of $Q_\mu^N$ and $Q_\mu^*$ have very similar proofs.

Version with $Q_\mu^N$

Let $\mathcal{T}^{QL}$ denote the backup operation in $Q$-Learning. Let $(\mathcal{T}^{QL})^m = \underbrace{\mathcal{T}^{QL} \circ \mathcal{T}^{QL} \circ \ldots \circ \mathcal{T}^{QL}}_{m \text{ times}}$. We know the following statements to be true:

$$Q_\mu^N = \mathcal{T}_\mu^N Q_\mu^N = r(s,a) + \gamma \cdot \mathbb{E}_{s'} \mathbb{E}_{\{a_i\}^N \sim \mu(a'|s')}[\max_{\{a_i\}^N} Q_\mu^N(s',a')] \tag{31}$$

$$\mathcal{T}^{QL} Q_\mu^N = r(s,a) + \gamma \cdot \mathbb{E}_{s'} \max_{a'} Q_\mu^N(s',a') \tag{32}$$

$$\lim_{m \to \infty} (\mathcal{T}^{QL})^m Q_\mu^N = Q^* \tag{33}$$

$$\left\| (\mathcal{T}^{QL})^{m+2} Q_\mu^N - (\mathcal{T}^{QL})^{m+1} Q_\mu^N \right\|_\infty \leq \gamma \cdot \left\| (\mathcal{T}^{QL})^{m+1} Q_\mu^N - (\mathcal{T}^{QL})^m Q_\mu^N \right\|_\infty \tag{34}$$

$$\left\| (\mathcal{T}^{QL})^{m+1} Q_\mu^N - (\mathcal{T}^{QL})^m Q_\mu^N \right\|_\infty \leq \gamma^m \cdot \left\| \mathcal{T}^{QL} Q_\mu^N - Q_\mu^N \right\|_\infty \tag{35}$$

Putting these together we have that,

$$\left\| Q_\mu^N - Q^* \right\|_\infty \leq \sum_{m=0}^\infty \left\| (\mathcal{T}^{QL})^{m+1} Q_\mu^N - (\mathcal{T}^{QL})^m Q_\mu^N \right\|_\infty \tag{36}$$

$$\leq \sum_{m=0}^\infty \gamma^m \cdot \left\| \mathcal{T}^{QL} Q_\mu^N - Q_\mu^N \right\|_\infty \tag{37}$$

$$= \frac{1}{1-\gamma} \left\| \mathcal{T}^{QL} Q_\mu^N - Q_\mu^N \right\|_\infty \tag{38}$$

$$= \frac{1}{1-\gamma} \max_{s,a} \left| \left( r(s,a) + \gamma \cdot \mathbb{E}_{s'} \max_{a'} Q_\mu^N(s',a') \right) \right. \tag{39}$$

$$\left. - \left( r(s,a) + \gamma \cdot \mathbb{E}_{s'} \mathbb{E}_{\{a_i\}^N \sim \mu(a'|s')}[\max_{\{a_i\}^N} Q_\mu^N(s',a')] \right) \right| \tag{40}$$

$$= \frac{\gamma}{1-\gamma} \max_{s,a} \left| \mathbb{E}_{s'} \left[ \max_{a'} Q_\mu^N(s',a') - \mathbb{E}_{\{a_i\}^N \sim \mu(a'|s')}[\max_{\{a_i\}^N} Q_\mu^N(s',a')] \right] \right| \tag{41}$$

$$\leq \frac{\gamma}{1-\gamma} \max_{s'} \left| \max_{a'} Q_\mu^N(s',a') - \mathbb{E}_{\{a_i\}^N \sim \mu(a'|s')}[\max_{\{a_i\}^N} Q_\mu^N(s',a')] \right| \tag{42}$$

Version with $Q_\mu^*$

Very similarly we have,

$$\left\| Q_\mu^N - Q^* \right\|_\infty \leq \sum_{m=0}^\infty \left\| (\mathcal{T}_\mu^N)^{m+1} Q^* - (\mathcal{T}_\mu^N)^m Q^* \right\|_\infty \tag{43}$$

$$\leq \sum_{m=0}^\infty \gamma^m \cdot \left\| \mathcal{T}_\mu^N Q^* - Q^* \right\|_\infty \tag{44}$$

$$= \frac{1}{1-\gamma} \left\| Q^* - \mathcal{T}_\mu^N Q^* \right\|_\infty \tag{45}$$

$$= \frac{1}{1-\gamma} \max_{s,a} \left| \left( r(s,a) + \gamma \cdot \mathbb{E}_{s'} \max_{a'} Q^*(s',a') \right) \right. \tag{46}$$

$$\left. - \left( r(s,a) + \gamma \cdot \mathbb{E}_{s'} \mathbb{E}_{\{a_i\}^N \sim \mu(a'|s')}[\max_{\{a_i\}^N} Q^*(s',a')] \right) \right| \tag{47}$$

$$= \frac{\gamma}{1-\gamma} \max_{s,a} \left| \mathbb{E}_{s'} \left[ \max_{a'} Q^*(s',a') - \mathbb{E}_{\{a_i\}^N \sim \mu(a'|s')}[\max_{\{a_i\}^N} Q^*(s',a')] \right] \right| \tag{48}$$

$$\leq \frac{\gamma}{1-\gamma} \max_{s'} \left| \max_{a'} Q^*(s',a') - \mathbb{E}_{\{a_i\}^N \sim \mu(a'|s')}[\max_{\{a_i\}^N} Q^*(s',a')] \right| \tag{49}$$

$$\square$$

**Corollary A.1.** *Let $V_\mu, Q_\mu, A_\mu$ denote the value, Q, and advantage functions of $\mu$ respectively. When $N = 1$ we have that,*

$$\|Q_\mu - Q^*\|_\infty \leq \frac{\gamma}{1-\gamma} \max_{s'} \left| \max_{a'} Q_\mu(s', a') - \mathbb{E}_{a' \sim \mu(a'|s')}[Q_\mu(s', a')] \right| \tag{50}$$

$$= \frac{\gamma}{1-\gamma} \max_{s'} \left| \max_{a'} Q_\mu(s', a') - V_\mu(s') \right| \tag{51}$$

$$= \frac{\gamma}{1-\gamma} \max_{s',a'} A_\mu(s', a') \tag{52}$$

*It is interesting how the sub-optimality can be upper-bounded in terms of a policy's own advantage function.*

**Corollary A.2.** *(Proof for second part of Theorem 3.3)*

*Proof.* We want to show $\lim_{N \to \infty} Q_\mu^N = Q^*$. More exactly, what we seek to show is the following,

$$\lim_{N \to \infty} \|Q_\mu^N - Q^*\|_\infty = 0 \tag{53}$$

or,

$$\forall \epsilon > 0, \exists N, \text{ s.t. } \forall M \geq N, \|Q_\mu^N - Q^*\|_\infty < \epsilon \tag{54}$$

Let $\epsilon > 0$. Recall,

$$\Delta(s) = \max_{a \in \text{Support}(\mu(\cdot|s))} Q_\mu^*(s, a) - \mathbb{E}_{\{a_i\}^N \sim \mu(\cdot|s)} [\max_{b \in \{a_i\}^N} Q_\mu^*(s, b)] \tag{55}$$

Let $\inf_s \mu^*(s) = p > 0$. Let the lower and upper bounds of rewards be $\ell$ and $L$, and let $\alpha = \frac{1}{1-\gamma}\ell$ and $\beta = \frac{1}{1-\gamma}L$. We have that,

$$\mathbb{E}_{\{a_i\}^N \sim \mu(\cdot|s)} [\max_{b \in \{a_i\}^N} Q_\mu^*(s, b)] \geq (1-p)^N \cdot \alpha + (1 - (1-p)^N) \cdot \max_{a \in \text{Support}(\mu(\cdot|s))} Q_\mu^*(s, a) \tag{56}$$

Hence $\forall s$,

$$\Delta(s) \leq (1-p)^N \cdot \max_{a \in \text{Support}(\mu(\cdot|s))} Q_\mu^*(s, a) - (1-p)^N \cdot \alpha \tag{57}$$

$$= (1-p)^N \cdot \left( \max_{a \in \text{Support}(\mu(\cdot|s))} Q_\mu^*(s, a) - \alpha \right) \tag{58}$$

$$\leq (1-p)^N \cdot \left( \beta - \alpha \right) \tag{59}$$

Thus, for large enough $N$ we have that,

$$\|Q_\mu^N - Q_\mu^*\|_\infty \leq \frac{\gamma}{1-\gamma} \max_s \Delta(s) < \epsilon \tag{60}$$

concluding the proof. □

# B AUTOREGRESSIVE GENERATIVE MODEL

The architecture for our autoregressive generative model is inspired by the works of (Metz et al., 2017; Van de Wiele et al., 2020; Germain et al., 2015). Given a state-action pair from the dataset $(s, a)$, first an MLP produces a $d$-dimensional embedding for $s$, which we will denote by $h$. Below, we use the notation $a_i$ to denote the $i^{th}$ index of $a$, and $a_{[:i]}$ to represent a slice from first up to and **not including** the $i^{th}$ index, where indexing begins at 0. We use a discretization in each action dimension. Thus, we discretize the range of each action dimension into $N$ uniformly sized bins, and represent $a$ by the labels of the bins. Let $\ell_i$ denote the label of the $i^{th}$ action index.

**Training** We use separate MLPs per action dimension. Each MLP takes in the $d$-dimensional state embedding and ground-truth actions before that index, and outputs $N$ logits for the choice over bins. The probability of a given index's label is given by,

$$p(\ell_i | s, a[:i]) = \text{SoftMax}\left( \text{MLP}_i(d, a[:i]) \right)[\ell_i] \tag{61}$$

We use standard maximum-likelihood training (i.e. cross-entropy loss).

**Sampling** Given a state $s$, to sample an action we again embed the state, and sample the action indices one-by-one.

$$p(\ell_0|s) = \text{SoftMax}\Big(\text{MLP}_i(d)\Big)[\ell_0] \tag{62}$$

$$\ell_0 \sim p(\ell_0|s), a_0 \sim \text{Uniform}(\text{Bin corresponding to } \ell_0) \tag{63}$$

$$p(\ell_i|s) = \text{SoftMax}\Big(\text{MLP}_i(d, a[:i])\Big)[\ell_i] \tag{64}$$

$$\ell_i \sim p(\ell_i|s, a[:i]), a_i \sim \text{Uniform}(\text{Bin corresponding to } \ell_i) \tag{65}$$

## C  ALGORITHM BOX

---

**Algorithm 2:** Full EMaQ Training Algorithm

---

Offline dataset $\mathcal{D}$, Pretrain $\mu(a|s)$ on $\mathcal{D}$

Initialize $K$ Q functions with parameters $\theta_i$, and $K$ target Q functions with parameters $\theta_i^{\text{target}}$
`Ensemble` parameter $\lambda$, Exponential moving average parameter $\alpha$

**Function** `Ensemble`(*values*)**:**
  **return** $\lambda \cdot \min(values) + (1 - \lambda) \cdot \max(values)$

**Function** $y_{\text{target}}$ $(s, a, s', r, t)$**:**
  $\{a_i'\}^N \sim \mu(a'|s')$
  $Qvalues \leftarrow [\ \ ]$
  **for** $k \leftarrow 1$ **to** $N$ **do**
    /* Estimate the value of action $a_k'$    */
    $Qvalues.\text{append}\Big(\text{Ensemble}\big([Q_i^{target}(s', a_k') \text{ for all } i]\big)\Big)$
  **return** $r + (1 - t) \cdot \gamma \max(Qvalues)$

**while** *not converged* **do**
  Sample a batch $\{(s_m, a_m, s_m', r_m, t_m)\}^M \sim \mathcal{D}$
  **for** $i = 1, ..., K$ **do**
    $\mathcal{L}(\theta_i) = \sum_m \Big(Q_i(s_m, a_m) - y_{\text{target}}(s_m, a_m, s_m', r_m, t_m)\Big)^2$
    $\theta_i \leftarrow \theta_i - \text{AdamUpdate}\Big(\mathcal{L}(\theta_i), \theta_i\Big)$
    $\theta_i^{\text{target}} \leftarrow \alpha \cdot \theta_i^{\text{target}} + (1 - \alpha) \cdot \theta_i$

---

## D  INCONCLUSIVE EXPERIMENTS

### D.1  UPDATING THE PROPOSAL DISTRIBUTION

Akin to the work of (Van de Wiele et al., 2020), we considered maintaining a second proposal distribution $\tilde{\mu}$ that is updated to distill $\arg\max_{\{a_i\}^N} Q(s, a)$, and sampling from the mixture of $\mu$ and $\tilde{\mu}$. In our experiments however, we did not observe noticeabel gains. This may potentially be due to the relative simplicity of the Mujoco benchmark domains, and may become more important in more challenging domains with more uniformly distributed $\mu(a|s)$.

## E  LAUNDRY LIST

- Autoregressive models are slow to generate samples from and EMaQ needs to take many samples, so it was slower to train than the alternative methods. However, this may be addressed by better generative models and engineering effort.

## F  ONLINE RL

EMaQ is also applicable to online RL setting. Combining strong offline RL methods with good exploration policies has the potential for producing highly sample-efficient online RL algorithms. Concretely, we refer to online RL as the setting where iteratively, a batch of $M$ environment steps with an exploration policy are interleaved with $M$ RL updates (Levine et al., 2020; Matsushima et al., 2020).

EMaQ is designed to remain within the support of the provided training distribution. This however, is problematic for online RL which requires good exploration interleaved with RL updates. To this end, first, we modify our autoregressive proposal distribution $\mu(a|s)$ by dividing the logits of all softmaxes by $\tau > 1$. This has the effect of smoothing the $\mu(a|s)$ distribution, and increasing the probability of sampling actions from the low-density regions and the boundaries of the support. Given this online proposal distribution, a criteria is required by which to choose amongst sampled actions. While there exists a rich literature on how to design effective RL exploration policies (Weng, 2020), in this work we used a simple UCB-style exploration criterion (Chen et al., 2017) as follows:

$$Q^{\text{explore}}(s, a) = \text{mean}\Big(\{Q_i(s, a)\}_K\Big) + \beta \cdot \text{std}\Big(\{Q_i(s, a)\}_K\Big) \qquad (66)$$

Given $N$ sampled actions from the modified proposal distribution, we take the action with highest $Q^{\text{explore}}$.

We compare the online variant of EMaQ with entropy-constrained Soft Actor Critic (SAC) with automatic tuning of the temperature parameter (Haarnoja et al., 2018). For EMaQ we swept the temperatures and used a fixed bin size of 40, 8 Q-function ensembles and $N = 200$. For fairness of comparisons, we also ran SAC with similar sweeps over different collection batch sizes and number of Q-function ensembles. In the fully online setting (trajectory batch size 1, Figure 3a), EMaQ is already competitive with SAC, and more excitingly, in the deployment-efficient setting[3] (trajectory batch size 50K, Figure 3b), EMaQ can outperform SAC[4]. Figures 4 and 5 present the results for all hyperparameter settings, for SAC and EMaQ, in the batch size 1 and batch size $50K$ settings respectively. **In the fully online setting, EMaQ is already competitive with SAC, and more excitingly, in the deployment-efficient setting, EMaQ can outperform SAC.**

## G  OFFLINE RL EXPERIMENTAL DETAILS

For each environment and data setting, we train an autoregressive model – as described above – on the provided data with 2 random seeds. These generative models are then frozen, and used by the downstream algorithms (EMaQ, BEAR, and BCQ) as the base behavior policy ($\mu(a|s)$ in EMaQ)[5].

### G.1  COMPARING OFFLINE RL METHODS

Following the bechmarking efforts of (Wu et al., 2019), the range of clipping factor considered for BCQ was $\Phi \in \{0.005, 0.015, 0.05, 0.15, 0.5\}$, and the range of target divergence value considered for BEAR was $\epsilon \in \{0.015, 0.05, 0.15, 0.5, 1.5\}$. For both methods, the larger the value of the hyperparameter is, the more the learned policy is allowed to deviate from the $\mu(a|s)$.

The rest of the hyperparameters use can be found in Table 2. The autoregressive models have the following architecture sizes (refer to Appendix B for description of the models used). The state embedding MLP consists of 2 hidden layers of dimension 750 with relu activations, followed by a linear embedding into a 750 dimensional state representation. The individual MLP for each action dimension consist of 3 hidden layers of dimension 256 with relu activations. Each action dimension is discretized into 40 equally sized bins.

---

[3]By deployment-efficient we mean that less number of different policies need to be executed in the environment, which may have substantial benefits for safety and otherwise constrained domains (Matsushima et al., 2020).

[4]It must be noted that the online variant of EMaQ has more hyperparameters to tune, and the relative performance is dependent on these hyperparameters, while SAC with ensembles has the one extra ensemble mixing parameter $\lambda$ to tune.

[5]While in the original presentation of BCQ and BEAR the behvior policy is learned online, there is technically no reason for this to be the case, and in theory both methods should benefit from this pretraining

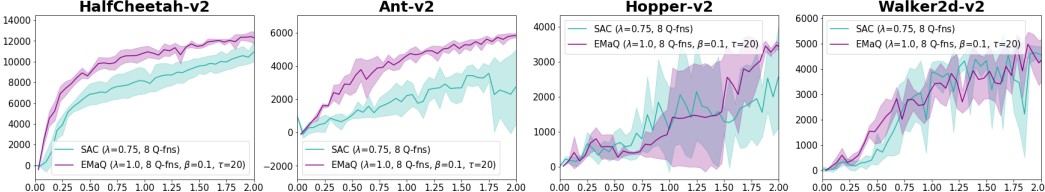

(a) SAC vs. EMaQ, Trajectory Batch Size 1: For easier visual interpretation we plot a single hyperparameter setting of EMaQ that tended to perform well across the 4 domains considered. The hyperparameters considered were $N = 200$, $\lambda = 1.0$, $\beta = 1.0$, $\tau \in \{1, 5, 10, 20\}$. SAC performed worse when using 8 Q-functions as in EMaQ. x-axis unit is 1 million environment steps.

(b) SAC vs. EMaQ, Trajectory Batch Size 50K: For easier visual interpretation we plot a single hyperparameter setting of EMaQ that tended to perform well across the 4 domains considered. The hyperparameters considered were $N = 200$, $\lambda \in \{0.75, 1.0\}$, $\beta \in \{0.1, 1.0\}$, $\tau \in \{1, 5, 10, 20\}$. x-axis unit is 1 million environment steps.

Figure 3: Online RL results under different trajectory batch sizes.

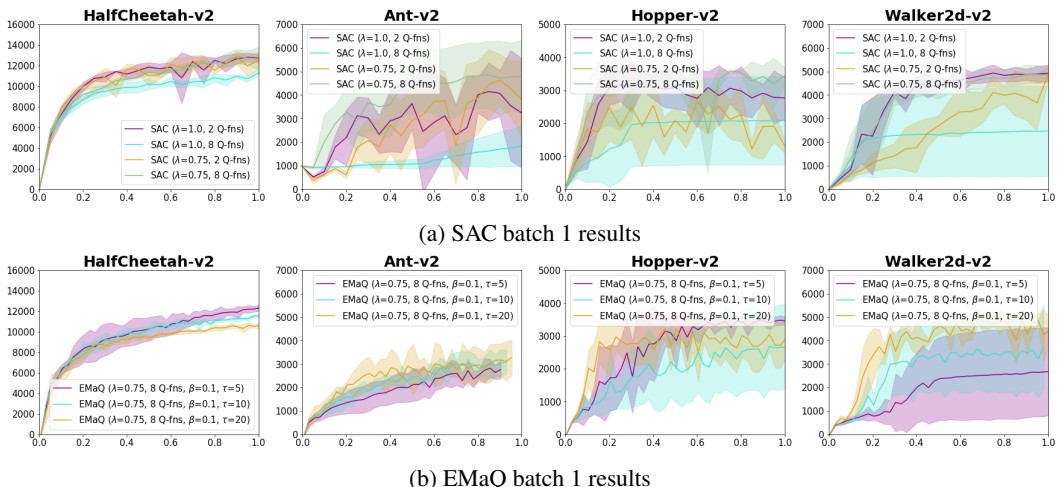

(a) SAC batch 1 results

(b) EMaQ batch 1 results

Figure 4: All results for batch size 1

## G.2 EMAQ ABLATION EXPERIMENT

Hyperparameters are identical to those in Table 2, except batch size is 100 and number of updates is 500K.

## G.3 DETAILS FOR TABLE **??** EXPERIMENTS

**Generative Model** The generative models used are almost identical to the description in Appendix B, with a slight modification that $\text{MLP}_i(d, a[: i])$ is replace with $\text{MLP}_i(d, \text{Lin}_i(a[: i]))$ where $\text{Lin}_i$ is a linear transformation. This change was not necessary for good performance; it was as architectural detail that we experimented with and did not revert prior generating Table **??**. The model dimensions for each domain are shown in 3 in the following format (state embedding MLP hidden size, state embedding MLP number of layers, action MLP hidden size, action MLP number of layers, Ouput

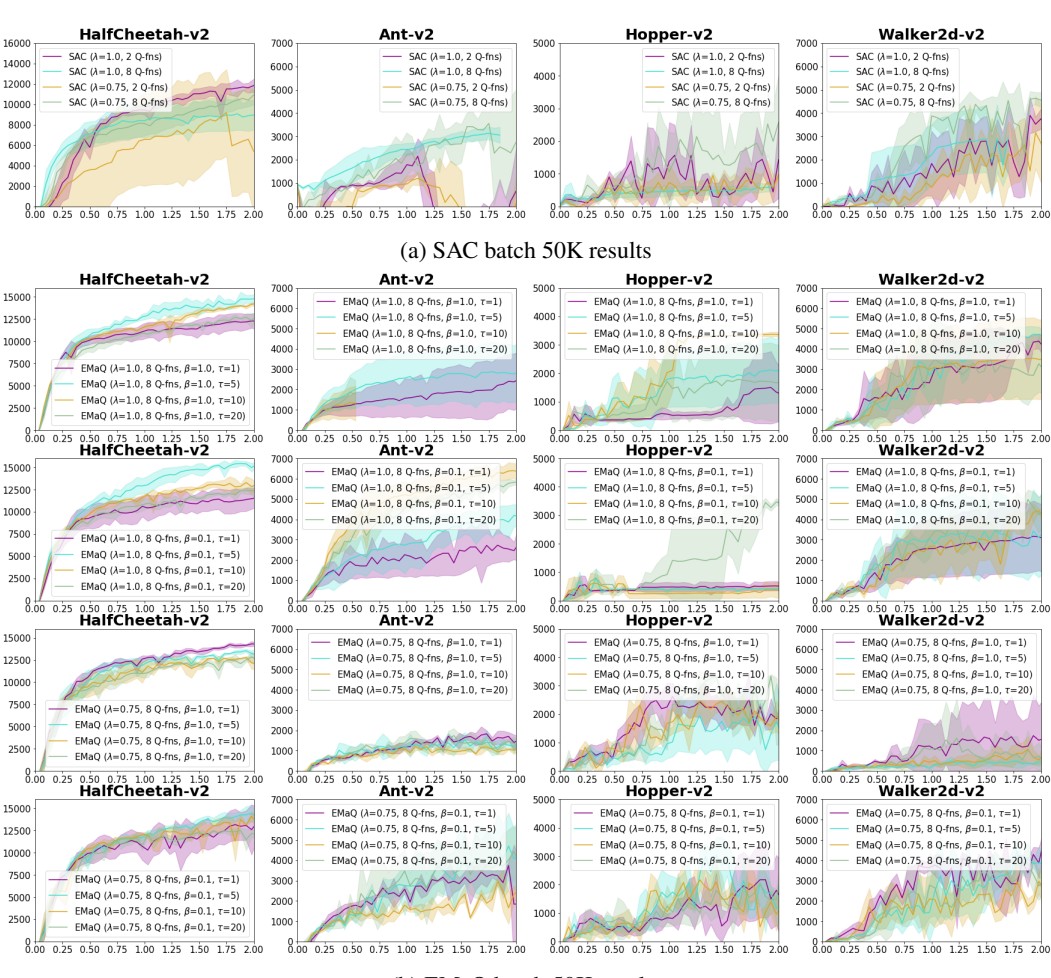

Figure 5: All results for batch size 50K

| Shared Hyperparameters | |
|---|---|
| $\lambda$ | 1.0 |
| Batch Size | 256 |
| Num Updates | 1e6 |
| Num $Q$ Functions | 8 |
| $Q$ Architecture | MLP, 3 layers, 750 hid dim, relu |
| $\mu$ lr | 5e-4 |
| $\alpha$ | 0.995 |
| **EMaQ Hyperparameters** | |
| $Q$ lr | 1e-4 |
| **BEAR Hyperparameters** | |
| $\pi$ Architecture | MLP, 3 layers, 750 hid dim, relu |
| $Q$ lr | 1e-3 |
| $\pi$ lr | 3e-5 |
| **BCQ Hyperparameters** | |
| $\pi$ Architecture | MLP, 3 layers, 750 hid dim, relu |
| $Q$ lr | 1e-4 |
| $\pi$ lr | 5e-4 |

Table 2: Hyperparameters for Mujoco Experiments

size of $\text{Lin}_i$, number of bins for action discretization). Increasing the number of discretization bins from 40 (value for standard Mujoco experiments) to 80 was the most important change. Output dimension of state-embedding MLP is the same as the hidden size.

**Hyperparameters** Table 3 shows the hyperparameters used for the experiments in Table **??**.

| Shared Hyperparameters | |
|---|---|
| $\lambda$ | 1.0 |
| Batch Size | 128 |
| Num Updates | 1e6 |
| Num $Q$ Functions | 16 |
| $Q$ Architecture | MLP, 4 layers, 256 hid dim, relu |
| $\alpha$ | 0.995 |
| $\mu$ lr | 5e-4 |
| Kitchen $\mu$ Arch Params | $(256, 4, 128, 1, 128, 80)$ |
| Antmaze $\mu$ Arch Params | $(256, 4, 128, 1, 128, 80)$ |
| Adroit $\mu$ Arch Params | $(256, 4, 128, 1, 128, 80)$ |
| **EMaQ Hyperparameters** | |
| $Q$ lr | 1e-4 |
| Kitchen N's Searched | $\{4, 8, 16, 32, 64\}$ |
| Antmaze N's Searched | $\{50, 100, 150, 200\}$ |
| Adroit N's Searched | $\{16, 32, 64, 128\}$ |
| **BEAR Hyperparameters** | |
| $\pi$ Architecture | MLP, 4 layers, 256 hid dim, relu |
| $Q$ lr | 1e-4 |
| $\pi$ lr | 5e-4 |
| **BCQ Hyperparameters** | |
| $\pi$ Architecture | MLP, 4 layers, 256 hid dim, relu |
| $Q$ lr | 1e-4 |
| $\pi$ lr | 5e-4 |

Table 3: Hyperparameters for Table 1 Experiments

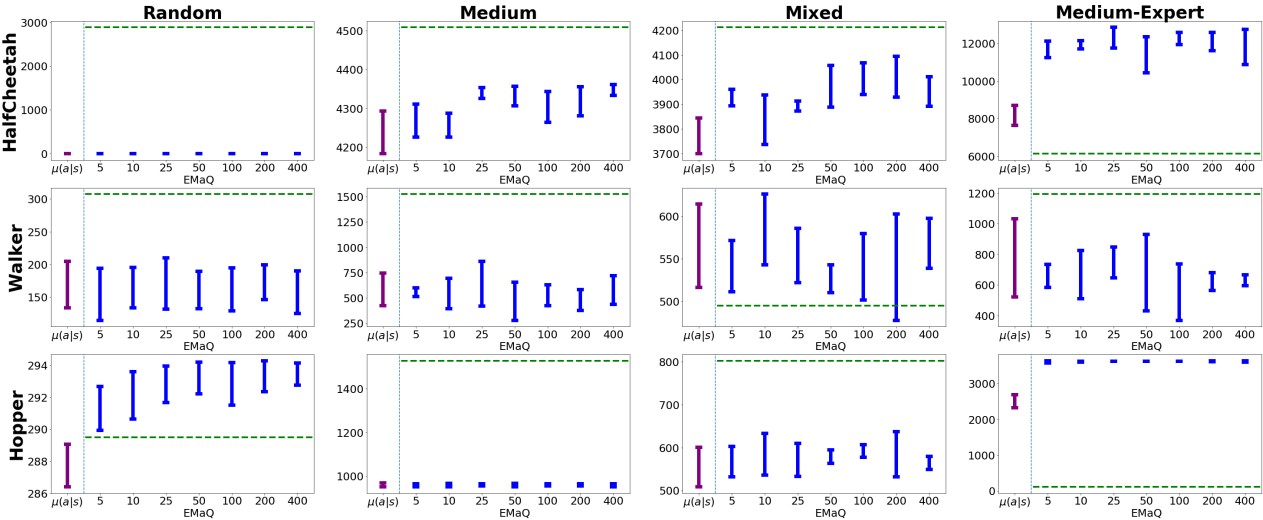

Figure 6: Results for evaluating EMaQ on D4RL (Fu et al., 2020b) benchmark domains when using the described VAE implementation, with $N \in \{5, 10, 25, 50, 100, 200, 400\}$. Values above $\mu(a|s)$ represent the result of evaluating the base behavior policies. Horizontal green lines represent the reported performance of BEAR in the D4RL benchmark (apples to apples comparisons in Figure 7).

## H  VAE RESULTS

### H.1  IMPLEMENTATION

We also ran experiments with VAE parameterizations for $\mu(a|s)$. To be approximately matched in parameter count with our autoregressive models, the encoder and decoder both have 3 hidden layers of size 1024 with relu activations. The dimension of the latent space was twice the number of action dimensions. The decoder outputs a vector $v$ which, and the decoder action distribution is defined to be $\mathcal{N}(\text{Tanh}(v), I)$. When sampling from the VAE, following prior work, samples from the VAE prior (spherical normal distribution) were clipped to the range $[-0.5, 0.5]$ and mean of the decoder distibution was used (i.e. the decoder distribution was not sampled from). The KL divergence loss term was weighted by 0.5. This VAE implementation was the one used in the benchmarking codebase of (Wu et al., 2019), so we did not modify it.

### H.2  RESULTS

As can be seen in Figure 6, EMaQ has a harder time improving upon $\mu(a|s)$ when using the VAE architecture described above. However, as can be seen in Figure 7, BCQ and BEAR do show some variability as well when switching to the VAEs. Since as an algorithm EMaQ is much more reliant on $\mu(a|s)$, our hypothesis is that if it is true that the autoregressive models better captured the action distribution, letting EMaQ not make poor generalizations to out-of-distribution actions. Figures 8 and 9 show autoregressive and VAE results side-by-side for easier comparison.

## I  EMAQ MEDIUM-EXPERT SETTING RESULTS

In HalfCheetah, increasing $N$ significantly slows down the convergence rate of the training curves; while large $N$s continue to improve, we were unable to train them long enough for convergence. In Walker, for EMaQ, BCQ, and most hyperparameter settings of BEAR, training curves have a prototypical shape of a hump, where performance improves up to a certain high value, and then continues to fall very low. In Hopper, for higher values of $N$ in EMaQ we observed that increasing batch size from 100 to 256 largely resolved the poor performance, but for consistency we did not alter Figure 1 with these values.

## J  COMPARISON WITH SOFTMAX BACKUP OPERATORS

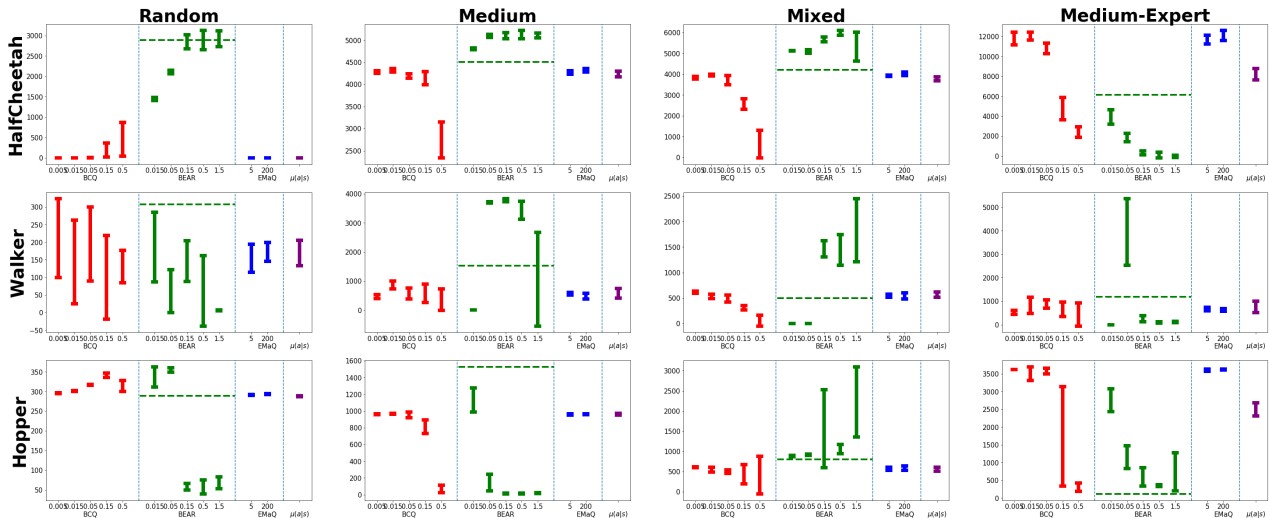

Figure 7: Comparison of EMaQ, BCQ, and BEAR on D4RL (Fu et al., 2020b) benchmark domains when using when using the described VAE implementation for $\mu(a|s)$. For both BCQ and BEAR, from left to right the allowed deviation from $\mu(a|s)$ increases. Horizontal green lines represent the reported performance of BEAR in the D4RL benchmark.

We thank one of our ICLR 2021 reviewers for the motivation for this section. For clarity of writing, we will write the forms for deterministic dynamics and remove the expectations over the next state.

An interesting connection to our proposed backup operators would be the following Softmax backup operator with similarities to EMaQ,

$$\mathcal{T}_\mu^\alpha Q(s,a) := r(s,a) + \mathbb{E}_{soft(a'|s')}[Q(s',a')] \tag{67}$$

$$soft(a|s) \propto \mu(a|s) \cdot \exp(\alpha \cdot Q(s,a)) \tag{68}$$

As suggested by our reviewer, the policy corresponding to $soft(a|s)$ is a policy that aims to maximize Q-values, subject to a KL-constraint between itself and the policy $\mu(a|s)$. The looser the constraint, the larger the effective $\alpha$ and the farther the policy will be from $\mu$. One approach to Monte Carlo estimation of the expectation on the right hand side could be to take samples using methods from the energy-based generative modelling literature.

An alternative approach which will more closely resembles EMaQ is to use self-normalized importance sampling,

$$\mathcal{T}_\mu^\alpha Q(s,a) := r(s,a) + \mathbb{E}_{soft(a'|s')}[Q(s',a')] \tag{69}$$

$$= r(s,a) + \sum_{\{a_i\}^N \sim \mu(a'|s')} w_i \cdot Q(s',a') \tag{70}$$

$$\tilde{w}_i = \frac{\mu(a_i'|s') \cdot \exp(\alpha \cdot Q(s',a_i'))}{\mu(a_i'|s')} \tag{71}$$

$$w_i = \frac{\tilde{w}_i}{\sum \tilde{w}_i} \tag{72}$$

$$= softmax(\alpha \cdot Q(s',a_i'))[i] \tag{73}$$

In this form, the soft backup is similar to EMaQ, where instead of taking ths max Q-value over the N samples, we take an average over the N Q-values, weighted by the softmax probabilities in equation 73. For a given N, the $\alpha = 0$ would be equivalent to Q-evaluation of the policy $\mu(a|s)$, and as $\alpha \to \infty$, the soft backups approach EMaQ backups.

In Figure 10 we present empirical results with the soft backup operators, under a large range $\alpha \in \{1, 4, 8, 16, 32, 64, 128, 256, 512, 1024\}$, in the Halfcheetah settings. The EMaQ and soft-EMaQ were run with the same architectures, but were smaller than the ones used for the results in the main text. We used the same checkpoints of the generative models as for the results in the main

text. The test-time policy for both approaches is the same, sampling $N$ actions and taking the argmax action under the ensemble Q-value. The only difference between the EMaQ and soft-EMaQ implementations was a one-line change to replace `max` with a softmax average of the Q-values.

Some interesting observations are the following: As anticipated, the soft EMaQ backups approach EMaQ as the value of $\alpha$ is increased. However, the necessary value of $\alpha$ to match the performance of EMaQ can be quite large. In the medium-expert setting, where figure 1 suggests challenges arising from the combination of large $N$s and function-approximators, we did not gain much advantage from soft backups, and only $\alpha \in \{8, 16, 32\}$ seem to have provided some mitigation of the problem for $N = 25$. Since the soft backup introduces an additional hyperparameter that cannot be determined ahead of time, and does not seem to provide an advantage (at least in the limited Halfcheetah settings considered), from a practical perspective, we would prefer to use the regular EMaQ backup.

## K  QUALITATIVE DIFFERENCES IN TRAINING CURVES

We have sometimes observed that the curves representing agent performance throughout training can be significantly more stable under EMaQ in comparison to BEAR and BCQ. A domain where the differences are particularly striking are the `antmaze-umaze` and `antmaze-umaze-diverse` domains. In figure 4 we have included plots of agent performance during training under the variety of considered hyperparameters and random seeds. It can be seen that in these two domains, initially the BCQ agents improves in performance close to the performance of EMaQ, and the drastically degrades with more training. In constrast, EMaQ agents remain stable even after twice as many training iterations as BCQ, which may indicate the downside of the heuristic perturbation model for constraining actions.

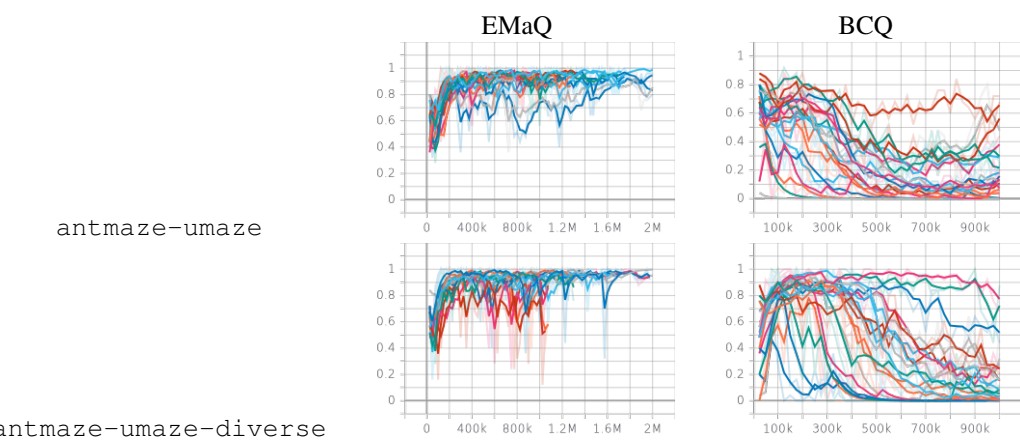

Table 4: Comparison of agent returns throughout training, under the variety of hyperparameters and and random seeds, in the small ant domains. We observe that EMaQ is significantly more stable than BCQ in these domains, even though the values of $N$ in EMaQ were fairly large for these plots $N \in \{50, 100, 150, 200\}$.

## L  LARGER PLOTS FOR VISIBILITY

Due to larger size of plots, each plot is shown on a separate page below. For ablation results, see Figure 11. For MuJoCo results, see Figure 12.

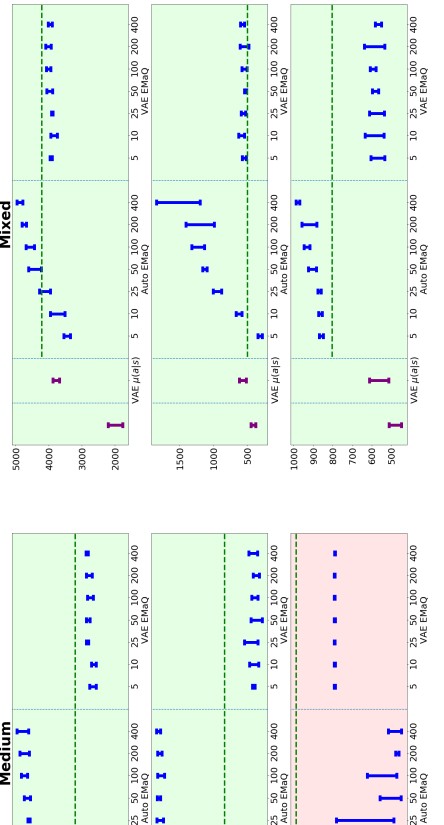

Figure 8: Results with both autoregressive and VAE models in one plot for easier comparison.

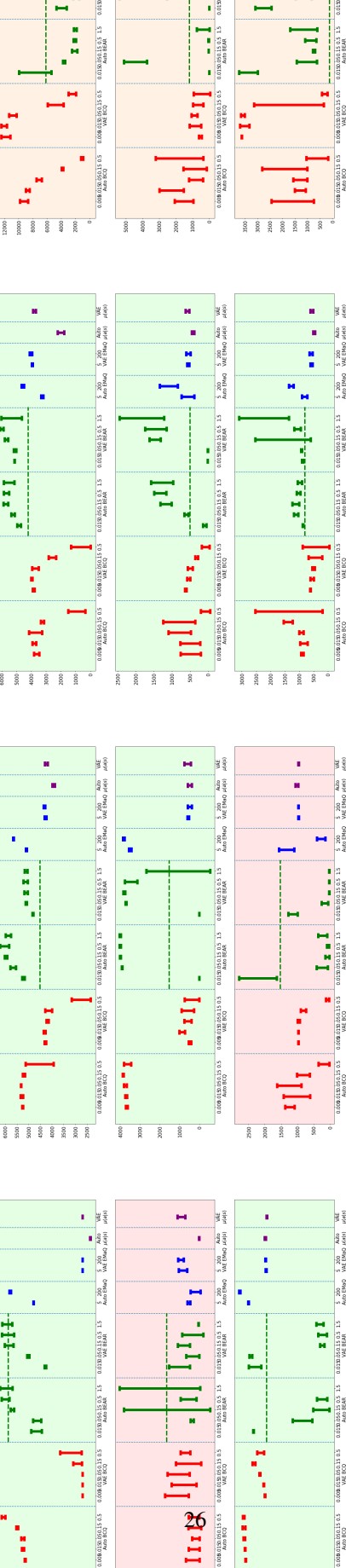

Figure 9: Results with both autoregressive and VAE models in one plot for easier comparison.

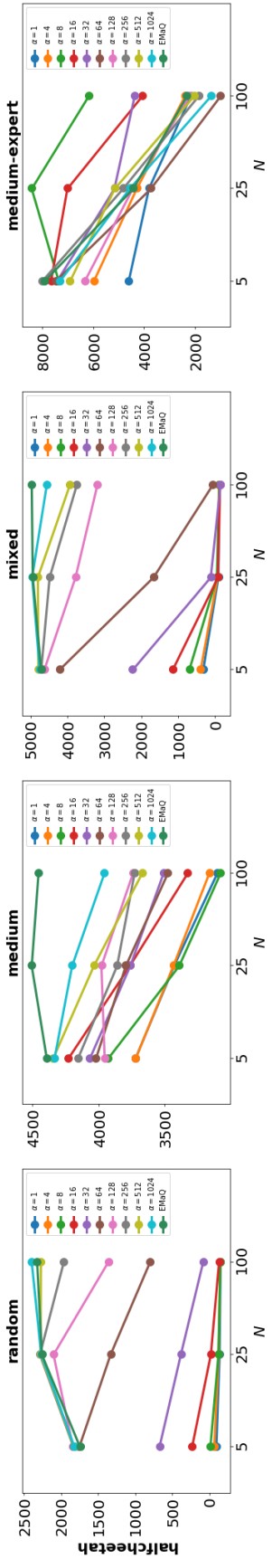

Figure 10: Comparison of Soft-EMaQ with EMaQ under a large range of hyperparameters in the Halfcheetah domain.

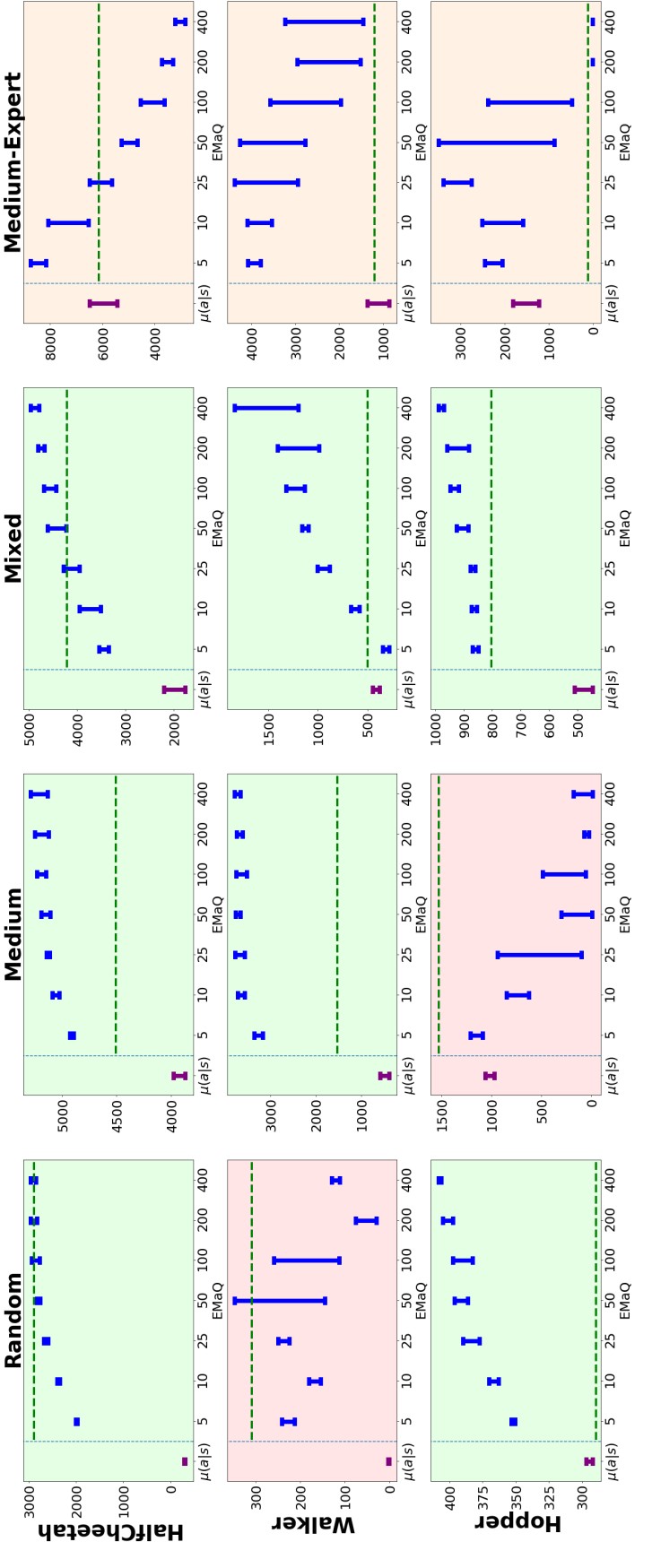

Figure 11: Results for evaluating EMaQ on D4RL (Fu et al., 2020b) benchmark domains, with $N \in \{5, 10, 25, 50, 100, 200, 400\}$. Values above $\mu(a|s)$ represent the result of evaluating the base behavior policies. Horizontal green lines represent the reported performance of BEAR in the D4RL benchmark (apples to apples comparisons in Figure 2). Refer to main text (Section 5.1) for description of color-coding.

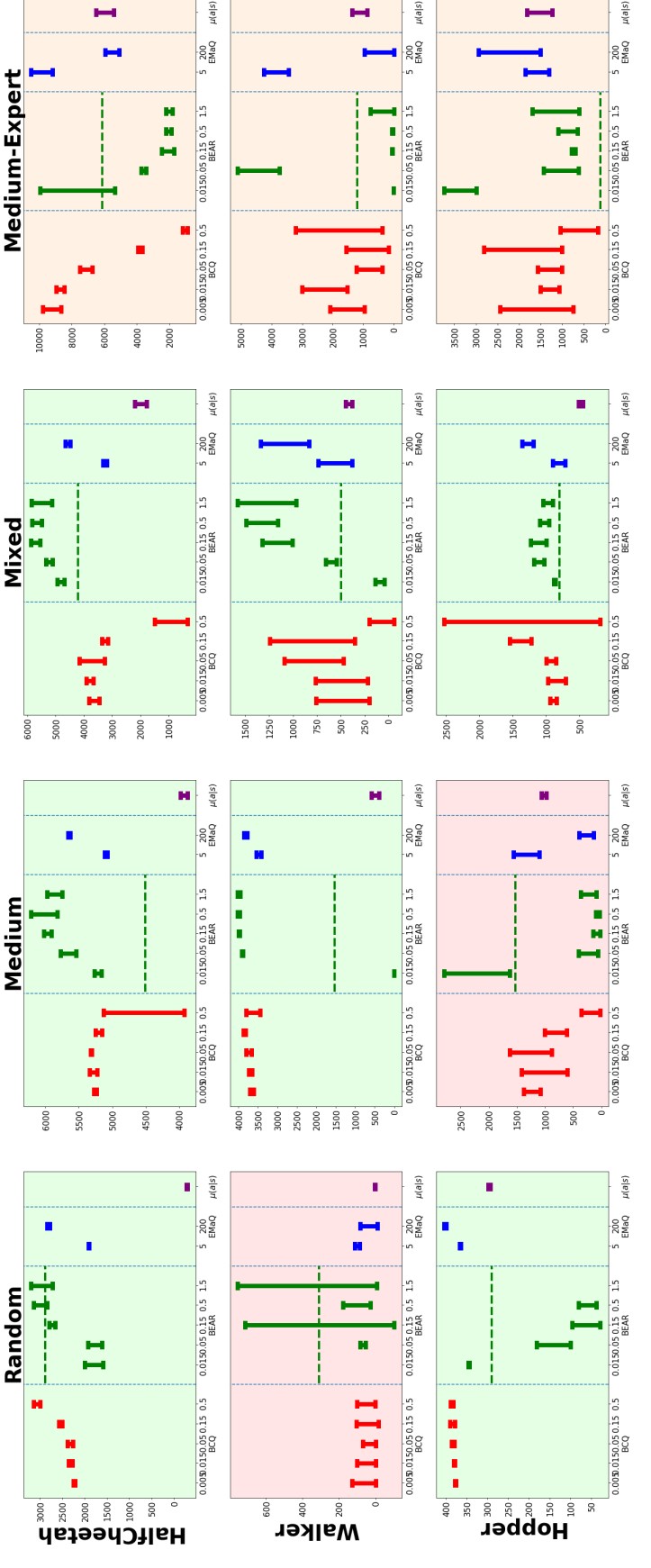

Figure 12: Comparison of EMaQ, BCQ, and BEAR on D4RL (Fu et al., 2020b) benchmark domains when using our proposed autoregressive $\mu(a|s)$. For both BCQ and BEAR, from left to right the allowed deviation from $\mu(a|s)$ increases. Horizontal green lines represent the reported performance of BEAR in the D4RL benchmark. Color-coding follows Figure 1.

