# OpenReview forum: "EMaQ: Expected-Max Q-Learning Operator for Simple Yet Effective Offline and Online RL"
_ICLR.cc/2021/Conference — Reject_

### Official Review · AnonReviewer2 · 2020-10-28
**An incremental algorithm changes compared to BCQ; Small empirical improvement**

**Rating:** 4
**Confidence:** 4

**Review:**

This paper proposed EMaQ, an approximation of Bellman operator, and thus yields a variant of deep Q learning algorithms in both online and offline settings. In tabular settings, several properties of the EMaQ operator such as the convergent guarantee, fixed point, and finite-sample error bounds were studied. Then the author shows the empirical performance of a deep learning approximation of the proposed algorithm on a standard batch RL benchmark. The authors also provide some analysis of the effect of different implementation choices on empirical performance, by ablation study.


A key fact is that the proposed method can be directly got from BCQ by settings the perturbation parameter to zero. Thus, the originality here is quite little. The proposed algorithm is a simplification compared with BCQ. However, in terms of both theoretical and empirical contributions, this paper does not show a significant improvement compared with BCQ. In general, the findings of this paper seem not surprising at all given BCQ and BEAR paper.

Pros:
1. Compared with BCQ, the proposed EMaQ operator has a simpler form with similar performance. In some sense, it is a Q learning analog of BCQ (which is actually an actor-critic algorithm).
2. There is some ablation study on the choice of generative model for $\mu(a|s)$, the number of action to sample, etc. which are missed in the original BCQ paper.

Cons:
1. As I mentioned in the summary, the biggest issue is that as a very simple and incremental algorithmic change, this paper also failed to provide additional discussion in either theory or experimental that can provide significantly more insight and inspiration to people. There are many potential directions where the discussion can go:
(1) In some sense it is a Q learning analog of BCQ (which is actually an actor-critic algorithm). It will be very interesting to discuss the effect of the difference between Q learning and actor-critic on BCQ v.s. EMaQ comparison. Unfortunately, I did not see enough in-depth discussion on that.
(2) The new algorithm is more closed the tabular settings. So does that yield stronger theoretical guarantees in function approximation settings? Or is there a conceptual experiment to demonstrate where the perturbation model is a bad idea/approximation and cause the failure of the algorithm?
(3) Does this very simple algorithmic choice lead to very different behavior of the algorithm (e.g. optimization path of the parameters, behavior of the resulting models other than just reward)? Why there is or is not a very different behavior of the algorithm?

2. According to the paper a key contribution of the empirical part is the importance of careful generative model design for
estimating behavior policies. This seems very straightforward as the BCQ/BEAR algorithm is using $\mu(a|s)$. Given that $\mu(a|s)$ is a plug-in module in those algorithms, new advances in the generative model can be immediately applied there so the significance and novelty of this finding are very limited.

3. A general problem of this paper is over-claiming. For example:
(1) "EMaQ matches and outperforms the prior state-of-the-art in the D4RL benchmarks" -- It matches (and did not outperform by any statistically significant margin) BCQ and BEAR. There are other baselines in D4RL that are better than BCQ/BEAR: BRAC, CQL, MOPO, MoRel. The former two are also model-free, and those numbers are all reported in the D4RL whitepaper.
(2) "EMaQ provides a quite intuitive and surprising measure of the complexity for offline RL problems." To claim that, I think it needs to show that  $\Delta$ is related to some general structure of the problem (theoretically) or some general phenomenon for a large group of algorithms (empirically).
These two claims can also be a potential improvement of this paper if they are really achieved.

---

> ### Author Response · Authors · 2020-11-25
> **Authors' Response (Part 1)**
>
> Thank you for your detailed review of our work. We hope that the following comments can address the comments you have raised.
>
> __Originality and Contribution__
>
> We agree that, as we have discussed in the main text, at the algorithmic level, EMaQ can be derived from BCQ by removing the perturbation network, a heuristic design choice for constraining the policy close to the base behavior policy. We believe that the key contributions of our work come from: 1) Studying this simplification in the Tabular setting, 2) Presenting strong empirical results despite the simplicity of the approach (more on this matter discussed below), and 3) bringing to light (theoretically and empirically) the notion of complexity based on the difference of max and sample-max (more on this discussed below), 4) Demonstrating performance equivalent to Soft-Actor-Critic in the online RL setting, with a very minor change in the algorithm (As highlighted by AnonReviewer1 To the best of our knowledge, no prior work has demonstrated state-of-the-art performance in both offline and online settings).
>
> __Additional Comments__
>
> We would be interested in further understanding your comment regarding the actor-critic vs. Q-learning. What aspect of this difference would you think is relevant for further studying and attention?
>
> __Where perturbation model would fail__
>
> An example of setting where the perturbation model be a bad approximation leading to failures is the following: Consider a simple point-mass maze domain. Imagine that in this domain, we have generated through some method many trajectories for the point-mass navigating from some arbitrary point A to an arbitrary point B. Now, we would like to use offline RL to make the agent navigate from a specific point to another target location. Notably, in the offline dataset, we can expect to have rarely generated actions that would push the point mass into the wall (for example, in a corridor, the offline data contains trajectories going back and forth, and won’t contain trajectories going left and right into the wall). In such a setting, a perturbation constraint is not a good idea: at a given spot in the maze, the perturbation along one axis should be high, and small in another. More importantly, at different points in the maze, the higher perturbation axis would change (depending on whether we are in a horizontal or vertical corridor). If in the RL backups we use a fixed perturbation, and sample actions with larger deviations in the incorrect direction, since there is no data in the offline setting, and because of function approximator inductive biases, the value function may for example think that it can pass through a wall. In terms of our empirical results, one domain which we believe can demonstrate this effect is the antmaze domain (whose results are included in Table 1 of updated manuscript, Table 3 in the original submission). __We have included a new appendix section K, discussing some training plots for these domains, comparing BCQ and EMaQ.__ Notably, we see that in the two antmaze-umaze domains, in BCQ, performance initially goes up close to the performance of EMaQ, and then drastically falls down, whereas we can see that with EMaQ, training is incredibly stable, even when we continue to train the model for twice as many iterations. Since the implementations of the two algorithms share many components, we believe that this is due to fundamental differences in the algorithmic choices.
>
> __Empirical Contribution__
>
> We agree that just changing the choice of generative model is not a novel contribution. We believe that the empirical contribution of our work is 1) demonstrating how significant the effect of this choice can be, and 2) showing that with this careful consideration, we can obtain very performant policies using a simplified algorithm (Table 1 in updated manuscript, Table 3 in original submission).
>
> __Contribution vs. Overclaiming__
>
> In terms of empirical results, in the kitchen and antmaze-umaze domains (Table 1 in updated manuscript, Table 3 in original submission)., EMaQ did outperform the prior approaches by a significant margin. Furthermore, to our knowledge, no prior work has reported such high numbers for these domains, including Conservative Q-Learning (CQL) which appears to be considered the most recent state-of-the-art algorithm (of course, reading the numbers from the CQL paper and comparing to our number is not an apples-to-apples comparison, but it is encouraging that we achieved higher numbers while using pessimistic ensembling instead of CQL’s Q-value regularization loss).

---

> > ### Author Response · Authors · 2020-11-25
> > **Authors' Response (Part 2)**
> >
> > __Inspiration from new notion of complexity__
> >
> > Aside from our empirical results, we believe that the introduced notion of complexity (Theorem 3.5) may serve as a source of inspiration for future work. Being able to approximate or derive proxies for the difference of max and expected sample-max, whether using $Q^*$ or $Q^N_\mu$, may be a valuable way to study the relative complexity of MDPs with similar state and action spaces. This “smoothness” phenomenon (difference of max and expected sample-max) is also presenting itself in some of our empirical results. As an example, in Figure 1, in many domains in the figure, small values of N perform significantly better than the base behavior policy. This suggests that the optimal Q function is quite flat with respect to a random policy’s action distribution, meaning that just a few samples are sufficient for sampling from high-value actions, with high probability.

---

### Official Review · AnonReviewer1 · 2020-10-28

**Rating:** 6
**Confidence:** 4

**Review:**

This paper analyzes off-policy algorithms using a sampling-based maximization of the Q function over a behavior policy for both Bellman targets and for policy execution, introducing the “Expected Max Q-learning” operator EMaQ for analysis. The paper then proposes to use more expressive autoregressive models (MADE) for learning the behavior policy from replay buffer data (either online or offline), which turns out to be very important, especially on harder tasks in the D4RL benchmark. The method is relatively novel - it is close to other methods in the literature, but they make and evaluate important design decisions that are valuable for the community. The results are quite strong, matching or exceeding BEAR on D4RL while at the same time matching SAC for online learning.

The proposed method is simple yet effective: they learn a behavior policy on offline data. Then they learn a Q-function by Bellman bootstrapping, using max of N samples from the behavior policy for the target step. Then for executing the policy, the method again uses the argmax of N samples from the behavior policy. As mentioned in the related work, prior work such as BCQ and QT-Opt has studied sample-based policies, and it has also been common to mix the two decisions: eg. BEAR uses a parametric policy for bootstrapping but then reports a boost from sampling for policy execution. But the proposed method is perhaps the simplest of these from an RL point of view, which is a good thing - and the juice really comes from the proposal to use MADE for modeling the behavior policy. This also suggests clear future directions for research - to explore other, better generative models (AR models in particular are probably a poor long-term solution because of slow sampling).

The analysis centers around the EMaQ operator, which is a nice representation for thinking about the maximization problem in Bellman bootstrapping. The paper proves the convergence of the operator in the tabular setting, and some intuitive theorems that larger values of N result in policies with better Q values (for policy evaluation), and thus better policy improvement. One issue though with N -> inf, is that it would tend to exploit function approximation errors of the Q function, and this is borne out in some of the results. Theoretical results along these lines would significantly strengthen the paper.

The highlight of the paper is results on both offline and online RL. In offline RL, EMaQ roughly matches the performance of BCQ and BEAR while being significantly simpler to implement (except perhaps the MADE model) and exceeds BCQ and BEAR in some environments and depending on the underlying implementation. In online RL, EMaQ roughly matches SAC, being slightly worse when updating the policy frequently and slightly better when updating the policy less frequently. As far as I know, no prior method has been able to show good performance on both: SAC is poor on offline training and BCQ/BEAR is poor on online training. This is the strongest positive for the paper.

Minor comments:

There are a couple of references to “number of samples” in the intro that do not make sense until reading the method. Perhaps you need to outline the method in the intro, instead of just hinting at it, if you want to discuss these details. “As an example, we observe that while a HalfCheetah random policy obtains a return of 0, a policy that at each state uniformly samples only 5 actions and chooses the one with the best value obtains a return of 2000.” - I’m not sure if I understood the setup here, or the significance. Is the value estimated with samples from the random behavior policy? Is the second policy you refer to your method (so fit a behavior policy on the random behavior data, then do Q learning + sample 5 actions for maximizing it), or something else? Moreover, I did not really understand why this is too surprising, since the bulk of the work is really being done by learning a good Q function and even on random-action data, learning a behavior policy would help you stay constrained to the data.

Deferring the full related work to the appendix skirts the page limit rules and is somewhat unfair to other submitters, and also made it hard to read since the links do not cross-reference between PDFs.

“Importantly, our theoretical analysis on the family of TD operators described by EMaQ apply beyond BCQ and can also provide new perspectives on some of the highly successful sample-max Q-learning algorithms (Kalashnikov et al., 2018a; Van de Wiele et al., 2020) – particularly on how the proposal distribution affects convergence.” Not sure if I missed this, but I didn’t exactly see such a result or know what this is referring to, which I would expect to be something like lower D(pi* || mu) implies faster convergence.

---

> ### Author Response · Authors · 2020-11-25
> **Authors' Response**
>
> Thank you for your review of our work. We hope that the following comments can address your questions.
>
> __Comment regarding the introduction__
>
> We agree with your comment that without explanation of the methodology, references to “number of samples” would not make sense. In the updated manuscript, we have marked with a red color the lines that we will remove.
>
> __Random Halfcheetah N=5 and why it is interesting__
>
> This is referring to EMaQ results on the HalfCheetah random setting, where with N=5 we observed very strong improvement over the base behavior estimate (2000 returns vs. 0). The reason we find this to be surprising is that the test-time policy is incredibly close to being just the base behavior policy: at each state, it is sampling 5 actions from the behavior policy and taking the best amongst these 5 under the learned Q-values. Hence, just 5 samples are “sufficient” to cover the action space well enough to produce a significantly better policy. From the perspective of Theorem 3.5, we believe that such empirical results demonstrate that many benchmark domains are “less complex” (in the sense of $\Delta(s,a)$) than one might intuitively expect, meaning that the sample-max of the optimal Q function, with few samples, closely approximates the full max of the optimal Q function.
>
> __Related Work__
>
> In light of your comments, we agree that Related Works should have been included in the main manuscript. To try to rectify the situation, using the additional page, we have replaced the shorter Related Works with the full version, and brought the results in Table 3 into the main text as Table 1 in case it may have been missed by reviewers.
>
> __Distribution affecting convergence__
>
> We agree that the phrasing in this scenario could have been improved. With this comment, we were referring to the result of Theorem 3.5 which highlights the difference between the max and the sample-max under a given $\mu$. We do not currently have theoretical results concerning convergence speed. We would very much appreciate pointers to works that in your view may have related theoretical results, so that we may be able to expand our theoretical investigations.

---

### Official Review · AnonReviewer4 · 2020-10-30
**The paper proposes a simple but effective algorithm.**

**Rating:** 6
**Confidence:** 3

**Review:**

Summary: The paper proposes the expected-max operator to address the problem of distribution shift in offline RL (which could also be applied in online RL). The paper establishes theoretical analysis of the proposed operator in convergence, sub-optimality, etc. A practical algorithm is proposed based on previous techniques (an ensemble of Q-functions) and a different generative model using an autoregressive architecture. Experiments demonstrate the effectiveness of EMaQ.

Strong points: The paper studies the important problem of how to mitigate distribution shift in offline RL, and proposes a simple but effective algorithm. The paper conducts extensive experiments comparing with state-of-the-art algorithms, with in-depth ablative studies of the effect of each component.

Concerns:
- I am surprised by the result in Figures 1 and 2 that EMaQ performs very well with a small value of N=5 (as the upper bound in Theorem 3.5 would be not very meaningful given a small N and gamma close 1). Could authors investigate the sampled 5 actions? Are they diverse enough to represent the entire action space well, considering the action space is continuous?

- The performance of EMaQ strongly relates to how good \mu is learned. What if \mu is close to a deterministic policy (e.g., \mu is not learned well and is close to a deterministic policy)? How to guarantee this diversity?

- Could authors compare the performance of EMaQ when N is 1 and an extremely large value, which could better justifies the performance in extreme cases.

- How are the computation time and learning curve (besides the final performance shown in the paper) of EMaQ compared with other methods?

- In Section 3.5, it is not clear how a small value of N could serve as a regularizer and smoothen the incorrect values.

---

> ### Author Response · Authors · 2020-11-25
> **Authors' Response**
>
> Thank you for your review of our work. We hope that the following comments can address your questions.
>
> __N=5__
>
> In each experiment, we first train the generative model on the offline dataset for the corresponding experiment, and use that as our $\mu(a|s)$. Hence, the extent to which the sampled actions are diverse is the extent to which the actions in the offline dataset are diverse. In the view of Theorem 3.5 and section 3.3, this suggests that the $\Delta(s,N)$ corresponding to these problems decreases very quickly as a function of N. Indeed, the empirical observation that N=5 works quite well in a number of cases is surprising, and we view this as suggesting that the optimal Q-functions for these specific settings may be “smooth” in the sense of Theorem 3.5.
>
> __Poor $\mu$__
>
> We would not expect this to work well since in our experiments, even with large VAEs, EMaQ was not able to perform well, which indicates the importance of a higher quality generative model. Rather, we believe our results demonstrate that not enough attention has been paid to the choice of generative models used, and with better choices, the upper bound of performance with a simple algorithm such as EMaQ is quite high, and matches and sometimes exceeds (Table 1 in the updated manuscript) prior state-of-the-art.
>
> __Very small and very large Ns__
>
> When N=1, EMaQ is equivalent to behavior cloning, whose results are included in Figures 1 and 2 (purple color, $\mu(a|s)$), and the results in Table 1 (same as Table 3 in the previous version of the manuscript) for other environments. The largest value we tried was N=400 in Figure 1 (and 200 in Figures 1 and 2) which is close to the largest we could fit on one GPU.
>
> __Computation Time__
>
> Autoregressive models are slower to sample from because each coordinate of the sample has to be sampled sequentially. However many recent advances in generative modeling (such as parallelized approximations to autoregressive models from the NLP literature) could address this shortcoming. Large number of action samples can slow down the method significantly. However, our empirical findings suggest that this value of N may not need to be very high practical settings (in the benchmark domains considered in this work).
>
> __Regularization effect of small N__
>
> When learning Q-functions, one could reasonably expect that the value estimates will be more accurate in higher density regions of the action distribution (density under the given offline dataset), and that erroneous overestimations occur in regions of the action distribution with less density (where samples are not observed in the offline dataset). With low values of $N$, the likelihood of $N$ samples containing actions from these low-density regions will be lower, and the expectation over the sample-maxes would “smoothen out” the effect of the low probability erroneous overestimations.

---

### Official Review · AnonReviewer5 · 2020-11-06
**Demonstrates the importance of how we model behavior policies in offline RL, but not sure of the significance of proposed algorithm**

**Rating:** 6
**Confidence:** 3

**Review:**

The paper proposes a simple offline RL algorithm based on Q-learning that instead of computing the exact maximum over actions, takes the sample maximum when sampling from an learned estimate of the behavior policy. Experimental results are quite promising (competitive with previous algorithms in both offline and online settings). The authors demonstrate that utilizing autoregressive models significantly improves over simple VAEs when used to model the behavior policy in offline learning (both for the proposed method as well as prior work), highlighting the need to place more focus on how we model the behavior policy in offline RL.

The article is clearly written, and the analysis of the proposed backup is original to the best of my knowledge.
However, with how important the choice of model used to model the behavior policy appears to be, I am unsure as to how significant the new proposed backup operator is. While I feel the paper clearly highlights the importance of how we model the behavior $\mu(a\vert s)$, I don't see particularly strong evidence, either theoretically or empirically, why one should prefer using the EMaQ backup.

Regarding the simplicity of the proposed algorithm compared to previous ones, EMaQ does use one fewer network since it does not need to explicitly model a separate policy. However, in order to achieve competitive performance, EMaQ did require a more complicated model for the behavior policy, and the increased sensitivity to that perhaps counteracts the benefits of not having an explicit policy. Additionally, EMaQ also needs to sample multiple actions for each backup and when executing the learned policy, incurring more computational costs in sampling compared to using an explicit policy network.

One thing to potentially examine more closely is how EMaQ and the regularization from choosing small $N$ relates to previously used forms of regularization. We could consider an alternate procedure where instead of sampling N samples from $\mu(a\vert s)$ and taking the max, we instead sample from the distribution $\mu(a\vert s) \exp(\alpha Q(s,a))$ (using something like self-normalized importance sampling for example) for some temperature $\alpha$. This corresponds to the policy that simultaneously maximizes the expected Q-value as well as  a KL divergence penalty against the original policy $\mu(a\vert s)$, and would resemble the sample maximum as the temperature increases. The key difference from other methods using KL-regularization against the behavior policy would be that it relies on samples solely from the behavior model, similarly to EMaQ. Comparing these two procedures with different parameters could perhaps give us a better understanding of how implicit regularization by choosing small $N$ in EMaQ relates to explicit regularization via KL.

**Complexity Measure**: I strongly disagree that $\Delta(s, N)$ is a useful notion of complexity for offline RL. While it captures a notion of how well the behavior policy covers good behaviors, it uses the exact $Q*$ or $Q_{\mu}^N$ values. However, the difficult part in offline RL is to accurately estimate those Q values, and the proposed complexity term has no way to account for sources of complexity such as stochasticity or the horizon of the MDP. For example, an MDP with only 2 actions per state can be potentially very complex, but all 2-action MDPs would presumably look trivial just judging by how fast $\Delta(s, N)$ decreases with N.

**Misc. Comments:**
With regards to the inductive and the choice of model class for the behavior policy, what if we did actually have access to the behavior policy that generated the data? Would EMaQ perform better with oracle access to the behavior policy, or is there something about behavior model learned from the actual transitions in the dataset that would allow for better learning? Perhaps such a question would be useful for examining what sorts of inductive biases should go into the behavior model.

**Updates after Author Response:**

*Complexity Measure*: My concern with the complexity measure is that it seems to me that the biggest difficulty in reinforcement learning is in actually being able to estimate accurate value functions, while proposed complexity measure really only vaguely captures how far optimal policies are from the behavior distribution.
In particular, at each state, it only depends on the true values of Q for each action, and can't capture how hard it is to estimate them.
Even among similarly structured MDPs, we can consider a 2-action MDP with a behavior policy that was simply uniform.
We could have one extremely simple MDP that was simply composed of independent deterministic bandit problems at each state with no transitions (always remain in your starting state), which would be trivial to solve even from offline data with full support.
On the other hand, we could have a much more complex MDP with meaningful stochastic transitions, random rewards and so on.
If we simply match the Q values in the two MDPs, they appear to be equally complex from this measure, despite the fact that the bandit MDP is far simpler to solve.
As such, I think the proposed complexity measure doesn't really reflect the real challenges of an offline RL problem, and am not sure how one would extend it to be useful.

*Re misc comments and access to true behavior policies:* My thinking here was that if we had a very stochastic behavior and finite samples, there would be actions that have reasonably high probability under the true behavior policy, but we never get to see in the data and wont' be able to evaluate well. One benefit of fitting the behavior model the the empirical data is that it would focus on those actions that do appear in the dataset (in an extreme case, we could simply have Dirac deltas on the observed data points), and so could benefit by restricting the actions to those that can be evaluated better.

*Overall Opinion:* In light of the empirical results mentioned in the author's response as well as the comparisons to KL regularization, I have raised my rating. I still do believe it is a very borderline paper, and would perhaps benefit from more careful analysis and focus on how the different behavior modelling choices influences offline.

---

> ### Author Response · Authors · 2020-11-25
> **Authors' Response (Part 1)**
>
> Thank you for your thorough feedback on our work. We hope that our response can address the comments you have brought to attention.
>
> __Evidence for why one should choose EMaQ__
>
> While from a practitioner's perspective many factors can influence the choice of algorithm, we believe that the main highlights of our approach are strong empirical performance with less moving parts and hyperparameters to tune. We would like to highlight that while EMaQ performs similar to BCQ and BEAR on the standard Mujoco locomotion tasks, in a number of domains such as the Kitchen tasks and the antmaze-umaze domains (Table 1 in updated manuscript, Table 3 in original submission), EMaQ substantially outperforms BCQ and BEAR. We believe that strong empirical performance despite simplicity could motivate the choice of EMaQ over other approaches.
>
> __Simplicity vs. Sensitivity__
>
> We believe that the more careful choice of generative model becomes a __necessity__ as we move towards more complex tasks, and all methods would require making similar considerations. As an example, we can compare to a baseline methods used in our work, BCQ. In BCQ, a perturbation network is used to modify samples from the behavior policy $\mu$ towards better actions, and the elementwise magnitude of the perturbation is capped with a hyperparameter $\Phi$. In our view, this is a fairly heuristic approach for constraining policies and introduces implicit assumptions about the nature of the problem domain which may not hold in more complex scenarios, as evidenced by results in Table 1 in the updated manuscript (Table 3 in the original submission). In our empirical results, we have demonstrated that removing such heuristic choices becomes feasible by focusing more on the choice of model.
>
> __KL regularization vs. EMaQ regularization__
>
> Thank you for your proposed comparison. We have included a new section in the __appendix (Section J)__, implementing your suggestion and ran experiments in the HalfCheetah domain, alongside discussion of methodology and results. From a conceptual perspective, for a given N, as a function of $\alpha$ the softmax backup interpolates between Q evaluation ($\alpha=0$) of $\mu$ and EMaQ ($\alpha \rightarrow \infty$). As a function of N, EMaQ interpolates between Q evaluation for $\mu$ and Q-learning. From a practical perspective, the challenge with softmax is that it is not clear how one would choose the $\alpha$ parameter effectively (as evidenced by results in Section J), especially since the magnitude of the Q-values will change during training.
>
> __Complexity Measure__ stochasticity and horizon are in the Q functions
>
> We agree that this measure is not practically useful in its current form, but our view was that it may be worth bringing to attention to influence potential directions for future theoretical work. Additionally, as noted, the bound holds while using $Q^*$ or $Q^\mu_N$ in the definition of $\Delta$, and depending on the setting and for smaller $N$, it may be possible to obtain somewhat accurate estimates.
> The quantity that this bound is bringing to attention is a new notion of “smoothness” for the optimal value functions, in the sense of difference of max and the expected sample-max. We believe this notion of complexity is intuitive (in contrast to bounds based on statistical divergences), and the smoothness phenomenon is presenting itself in some of our empirical results. As an example, in Figure 1, in many domains in the figure, small values of N perform significantly better than the base behavior policy. This suggests that the optimal Q function is quite flat with respect to a random policy’s action distribution, meaning that just a few samples are sufficient for sampling from high-value actions, with high probability.
> Regarding your comment on stochasticity and horizon, these notions would be captured within the Q-functions.
> Regarding your comment on 2-action MDPs, we would envision a proxy of $\Delta(s,a)$ to provide us with a relative notion of complexity, and not necessarily transferable amongst MDP with different state or action spaces.
> If any come to mind, we would be very interested in hearing any additional comments on this issue.

---

> > ### Author Response · Authors · 2020-11-25
> > **Authors' Response (Part 2)**
> >
> > __Misc Comments__
> >
> > We did not perform such an experiment since all our experiments were in the D4RL dataset and we did not have access to the policies that generated the datasets. However, we would imagine that this would only improve performance, since it would remove a source of estimation error; if we have access to the true behavior policy, all action samples will __definitely__ be coming from the correct action distribution that generated the offline dataset, reducing the extent to which we would be evaluating and optimizing Q-functions on out of distribution actions. Our results using the less-capable VAE models also provide evidence that being closer to the true behavior policy is better. Nonetheless, we find your question intriguing and were wondering if you had a scenario in mind where using the true behavior policy could be worse than the a learned behavior estimate?
> >
> > __Additional Comment__
> >
> > In addition to our response above, we would like to highlight our results in the online RL setting, where we matched the performance of soft-actor-critic using a slight modification of EMaQ. As noted by AnonReviewer1 To the best of our knowledge, no prior work has demonstrated state-of-the-art performance in both offline and online settings.

---

### Author Response · Authors · 2020-11-25
**Additional Comments Addressed to All Reviewers**

We thank all reviewers for the significant effort they have placed in providing us with high quality and insightful reviews.

__Comments We Would Like to Highlight__

Overall, we believe the key contributions of our work are the following:
1) Introducing a new backup operator, that results in a simplification of prior work algorithmically (BCQ), and enable closer theoretical examination in the Tabular setting
2) Presenting strong empirical results despite the simplicity of the approach, demonstrating the importance of more careful choice of generative models, and obtaining state-of-the-art in a number of D4RL domains (Table 1 in updated document, or Table 3 in the original submission)
3) Bringing to light (theoretically and with empirical observations) the notion of complexity based on the difference of max and expected sample-max to inspire future work in this direction
4) Demonstrating performance equivalent to Soft-Actor-Critic in the online RL setting, with a very minor change in the algorithm. As highlighted by AnonReviewer1 To the best of our knowledge, no prior work has demonstrated state-of-the-art performance in both offline and online settings.

__Important Manuscript Updates__

* In the updated revision, __we have not split the document into a main text and appendix__, and included all section in one file for easier navigation.
* The red color in the Introduction indicates sentences that will be removed.
* The short Related Works section has been replaced with the full version from the appendix.
* __We have brought Table 3 into the main text as Table 1__, since we believe some reviewers may have not noticed these strong empirical results on additional D4RL domains.
* __We have added two new sections to the appendix (J and K)__ for addressing some comments for the reviewers, which may be relevant for all reviewers.

---

### Decision · Program_Chairs · 2021-01-07
**Final Decision**

**Decision:**

Reject

**Comment:**

**Overview** The paper provides a simplified offline RL algorithm based on BCQ. It analyzes the algorithms using a sampling-based maximization of the Q function over a behavior policy for both Bellman targets and for policy execution -- the EMaQ. Based on this, the paper then proposes to use more expressive autoregressive models (MADE) for learning the behavior policy from replay buffer data. The methods work well for harder tasks in the D4RL benchmark.

**Pro**
- The method is relatively novel
- Algorithms are simple modifications of existing ones
- Empirical results are strong, matching or exceeding BEAR on D4RL while at the same time matching SAC for online learning
- Work for both and offline
- ablation study on the choice of generative model for μ(a|s)

**Con**
- The current form of the complexity measure is somewhat not practical.
- Theoretical results are not strong enough
- Algorithmic contributions appear incremental

**Recommendation** The paper is on the borderline. It contributes simple and nice algorithmic ideas and these ideas work well empirically. These results demonstrate that a good choice of the behavior policy generative model is important for some tasks. At the same time, the reviewers are concerned about the theoretical parts, e.g., issues relates new complexity measure. Overall, the meta-reviewer believes that the paper might not be in a status ready for publication.